# Long-Term Antibacterial Effect of Electrospun Polyvinyl Alcohol/Polyacrylate Sodium Nanofiber Containing Nisin-Loaded Nanoparticles

**DOI:** 10.3390/nano10091803

**Published:** 2020-09-10

**Authors:** Yue Jiang, Donghui Ma, Tengteng Ji, Dur E Sameen, Saeed Ahmed, Suqing Li, Yaowen Liu

**Affiliations:** 1College of Food Science, Sichuan Agricultural University, Ya’an 625014, China; jiangyue@stu.sicau.edu.cn (Y.J.); 18894314357@163.com (D.M.); jtt5881@163.com (T.J.); sameen0388@gmail.com (D.E.S.); saeedahmedM1993@gmail.com (S.A.); 2School of Materials Science and Engineering, Southwest Jiaotong University, Chengdu 610031, China

**Keywords:** nisin, encapsulation, ultrasonic, polyvinyl alcohol, polyacrylate sodium, nanofibers

## Abstract

Response Surface Methodology (RSM) was used to assess the optimal conditions for a Water/Oil/Water (W/O/W) emulsion for encapsulated nisin (EN). Nano-encapsulated nisin had high encapsulation efficiencies (EE) (86.66 ± 1.59%), small particle size (320 ± 20 nm), and low polydispersity index (0.27). Biodegradable polyvinyl alcohol (PVA) and polyacrylate sodium (PAAS) were blended with EN and prepared by electrospinning. Scanning electron microscopy (SEM) revealed PVA/PAAS/EN nanofibers with good morphology, and that their EN activity and mechanical properties were enhanced. When the ultrasonication time was 15 min and 15% EN was added, the nanofibers had optimal mechanical, light transmittance, and barrier properties. Besides, the release behavior of nisin from the nanofibers fit the Korsemeyer–Peppas (KP) model, a maximum nisin release rate of 85.28 ± 2.38% was achieved over 16 days. At 4 °C, the growth of *Escherichia coli* and *Staphylococcus aureus* was inhibited for 16 days in nanofibers under different ultrasonic times. The application of the fiber in food packaging can effectively inhibit the activity of food microorganisms and prolong the shelf life of strawberries, displaying a great potential application for food preservation.

## 1. Introduction

Nowadays, microbial pollution has become a major global public health problem. It is an important research direction to add antibacterial agents into basic materials to prepare new materials with antibacterial effects. In the food industry, chemical antimicrobial agents are usually used to inhibit the growth of microorganisms and prolong the shelf life of food. However, in recent years, due to the widespread use of chemical antimicrobial agents, food safety problems and human health risks have become a concern [1]. Metal antimicrobial agents (such as silver, copper, zinc, and nickel) can show a broad antibacterial spectrum and effectively inhibit microbial growth, but high levels of metal elements significantly increase the risk of poisoning [2]. Natural antibacterial agents are the development direction of antibacterial agents in the future. However, essential oils have been widely used as food preservatives. Although essential oils have unique properties, their applications have been limited due to their low antibacterial activity, chemical complexity, and strong odor [3]. Bacteriocin has attracted much attention as a antimicrobial agent due to its natural, safe and spectrum antibacterial property. Bacteriocins are toxins produced by most bacteria and can kill or block the growth of other closely related bacteria in order to compete for nutrients and space in a limited environment. Most bacteriocins are active only against closely related bacteria and thus exhibit targeted, narrow spectrum activity. This leaves the beneficial bacteria unaffected, further reducing the risk of developing antimicrobial resistance [4]. Nisin has been granted use as a preservative in the food industry, and is a bacterial antibacterial peptide produced by the food grade lactic acid bacterium named *Lactococcus lactis subsp. Lactis*. It has broad-spectrum efficacy against Gram-positive bacteria and is extensively used in the food industry. However, the application of nisin is limited as it is unstable and insoluble [5]. A microcapsule system based on liposomes and polysaccharides is an important way to solve this problem, which has been developed. Wu et al. prepared and characterized chitosan-nisin (CS-nisin) microcapsules [6]. Relative to CS or nisin alone, CS-nisin microcapsules were significantly more effective at inhibiting microbial growth, lipid peroxidation, and protein degradation. Amara et al. tried complexation to encapsulate nisin (5 g L^−1^) via a spray-drying technique [7]. Complexation with pectin or alginate preserved nisin structure and antimicrobial activity during spray-drying. Maresca et al. prepared and characterized alginate-nisin microcapsules [8]. In foods stored at 4 °C and with a pH of 4.5 or 6.0, microcapsule antimicrobial activity was effectively preserved. Hassan et al. developed an antimicrobial alginate/resistant starch microcapsule containing nisin [9]. Its highest encapsulation efficiency (EE) was 33% and it created an inhibition zone of 15 ± 2 mm against Pediococcus acidilactici UL5 after 18 h. However, numerous factors may affect microcapsule properties. Stable nisin microcapsules with excellent antimicrobial performance were prepared by response surface methodology (RSM) based on a Box–Behnken design. This process is simple to operate, requires few experimental runs for analysis, and optimizes microcapsule preparation. Hu et al. used RSM to optimize CS-nisin microcapsule preparation [10]. The ideal conditions were CS = 2.4 mg mL^−1^, salt addition at 8 mL min^−1^, 3.8:1 (*w*/*w*) CS: nisin ratio, and Na_2_SO_4_ precipitant. Further, 1% CS-nisin had maximum antibacterial activity at pH 5.0–6.0 and created an inhibition zone 19.85 ± 1.31 mm in diameter against Bacillus subtilis.

Gelatin is a biodegradable protein material with excellent water solubility, emulsification and, thickening capabilities, along with high crosslinking activity. It is produced by the partial hydrolysis of collagen, which is still the main commercial choice for wall materials [11]. De Souza et al. studied gelatin and five different polysaccharides, including Arabic gum and pectin, to encapsulate cinnamomum zeylanicum by composite coacervation. The particles obtained from different materials have high entrapment efficiency. The microencapsulation process maintains the bioactivity potential of cinnamon extract and conceals the undesirable sensory properties, so that it can be used as a functional component in food and as a health care product [12]. In the work of Oliveira et al., green coffee oil loaded with caffeine and kawasol was encapsulated with cashew gum and gelatin to prepare microcapsules. The particle with 25% green coffee oil had good encapsulation efficiency (85.57%). The microcapsules were stable under the processing conditions of tamarind juice, and were able to be mixed into the juice without changing its rheological or sensory properties and remained stable during storage for 30 days [13]. The coacervates were able to encapsulate the lipid extract (astaxanthin encapsulation efficiency 59.9 ± 0.01%), forming multinucleated, polymorphic microcapsules with an average size of 32.7 ± 9.7 μm by the gelatin and cashew gum. Microcapsules are well dispersed in pure yogurt, which can improve the coloring ability, although no differences in odor are found [14]. Several studies reported that the microcapsules prepared by gelatin encapsulation were successfully applied in the food industry. The microencapsulation of nisin prepared by gelatin encapsulation can effectively protect nisin and has broad development prospects in the food industry. The use of polymers such as polyvinyl alcohol (PVA) has attracted interest, as they are biocompatible, biodegradable, nontoxic, chemically resistant, and film-forming. They are used as environmentally friendly packaging film material [15]. Polyacrylate sodium (PAAS) has been used in biomedical fields as a superabsorbent, self-healing medical implant because of its good material properties and biocompatibility [16]. It is nontoxic, durable, cost-effective, and biodegradable. Polymer based nanofibers are considered as potential materials in a wide range of fields due to their excellent properties, such as high specific surface area and easy functionalization. Electrospinning technology, including multi axis electrospinning, is a general method for the production of the fiber membranes of various natural and synthetic materials [4]. Electrospinning has been investigated for protective textiles, biomedical and food packaging as it generates nanofibers with properties not found in traditional fibers such as high pore interconnectivity, high specific surface area, surface functionalization, excellent breathability, tunable porosity, and easy manipulation of chemical compositions and structures for desired properties and functionalities [17]. Previous studies demonstrated that inorganic salts help generate fibers of uniform size and markedly reduce the number and size of beaded structures that form on them [18]. Hence, combining PVA and PAAS by electrospinning could yield homogeneous fibers with excellent morphology. Aminyan et al. prepared superabsorbent nanofibers from polyacrylic acid (PAA) and NaOH by electrospinning and investigated product swelling performance [19]. The nanofibers had ≤ 90,000% water swelling ratios. Jin et al. prepared fibrous membranes and cast films from aqueous mixtures of PVA and PAA via electrospinning and cast solution, respectively [20]. Membrane and film swelling increased with pH but the swollen fibrous membranes were dramatically stronger and more absorbent than the cast films. Some studies have indicated that nisin may be incorporated into nanofibers through electrospinning. Soto et al. prepared biodegradable antimicrobial nanofibers based on amaranth protein isolate: pullulan (API: PUL) plus nisin [21]. The nisin in the (API: PUL) nanofibers had a release rate of 81.49% at pH 3.4 after 12 h. When nisin API:PUL fibers were applied to complete bactericidal activity against *Salmonella Typhimurium, L. monocytogenes* and *L. mesenteroides* inoculated in fresh cheese, microorganism inactivation was complete after 142, 120, and 170 h, respectively. Cui et al. reported that polyethylene oxide nanofibers embedded with nisin-loaded poly-γ-glutamic acid/chitosan nanoparticles were highly efficacious against *L. monocytogenes* [22]. The *L. monocytogenes* density on fresh cheese decreased from 3.19 log CFU g^−1^ to 1.43 log CFU g^−1^ after 7 d. Thus, it can be seen that the nanofiber film embedded with nisin nanoparticles has a development prospect in the application of active packaging for food preservation. Han et al. prepared triaxial fiber membranes containing nisin by electrospinning [4]. Their antimicrobial activity persisted for 7 d. In the first 5 d, the membranes killed > 99.99% of the *S. aureus* cells and were superior to the other types of membranes tested.

High-energy methods such as ultrasonication have been used to form microcapsules and fibrous membranes because ultrasonic agitation is far stronger in solution than mechanical agitation. Ultrasonication also generates and disperses particles that are homogeneous in size, highly stable, and that have low polydispersity indices in solution [23]. Wang et al. found that ultrasonication accelerates crystallization and improves particle size and morphology [24]. Liu et al. reported that ultrasonication improves the structure, light transmittance, and mechanical and moisture barrier properties of film surfaces relative to those of untreated films [25].

To the best of our knowledge, no studies have attempted to use RSM to optimize encapsulated nisin (EN) with maximum EE or prepare antimicrobial PVA/PAAS nanofibers containing EN. A combination of EN and PVA/PAAS nanofibers could effectively protect and control the release of nisin. Here, we prepared stable EN with excellent antimicrobial performance. We fabricated PVA/PAAS nanofibers containing EN by electrospinning and optimized them by ultrasonication. The nanofiber product had high pore interconnectivity, specific surface area, and surface functionalization. Ultrasonication facilitates EN dispersion, PVA and PAAS blending, and the formation of nanofibers with homogeneous diameters by electrospinning. We determined the optimal nanofiber-forming quantities of PVA/PAAS/EN and established the influence of ultrasonication time on nanofiber functionality, and at the same time, analyzed the release behavior of nisin from nanofibers. Thus, the aim of this work is to develop a new antibacterial nanofiber material by electrospinning technology, which is doped with nisin nanoparticles, and is expected to be used as active food packaging.

## 2. Materials and Methods 

### 2.1. Materials

PVA (degree of polymerization, 1799 ± 50; degree of deacetylation, ≥ 97.0%) and PAAS (average MW, 4.0–5.0 × 10^6^) were purchased from Chengdu Kelong Chemicals Co. Ltd. (Chengdu, China). Nisin, gelatin, and soy oil were obtained from Shanghai Xinglong Biotechnology Co. Ltd. (Shanghai, China). *Staphylococcus aureus* (*S. aureus*) *ATCC* 29523 and *Escherichia coli* (*E. Coli*) *ATCC* 25922 were acquired in lyophilized form from the CICC (China Center of Industrial Culture Collection, Beijing, China).

### 2.2. Encapsulated Nisin Preparation

W/O/W emulsions were fabricated according to the method used by Huang et al., with certain modifications [26]. First, 1 g of 0.1 g mL^−1^ nisin was dissolved in distilled water (inner water phase) and placed in 20 mL soy oil (oil phase) containing various mount of Stepan^®^ 80 (E_1_). Second, the mixture was placed in 2.5 g of 1% (*w*/*v*) gelatin (outer phase) containing Tween^®^ 80 (E_2_) and stirred for 10 min at 25 °C. Third, the pH of each mixed emulsion was adjusted to 3, 4, and 5, respectively, using HCl (1 mM) or NaOH (1 mM). Ultrasonication was applied for 0 min, 2.5 min, and 5 min at 43 kHz and 200 W (Table 1). EN was obtained by centrifugation at 6000 rpm and 4 °C for 5 min followed by freeze-drying at −50 °C for 48 h.

### 2.3. Encapsulated Nisin Optimization by Response Surface Methodology

The Box–Behnken (BBD) response surface methodology was used to optimize the formulation for maximum EN efficiency. The effects of E_1_/E_2_ (X_1_), pH (X_2_), and ultrasonication time (X_3_) and their interactions on encapsulation efficiency were evaluated. Appendix A shows the actual and coded levels of each factor in terms of BBD. A second-order polynomial equation was used to identify the predictable response:
(1)$$Y=A_{0}+3\sum_{i=1}^{3}(A_{i}X_{i})+\sum_{i=1}^{3}A_{\mathrm{ii}}X_{i}^{2}+\sum_{i=1}^{2}\sum_{j=i+1}^{3}A_{\mathrm{ii}}X_{\mathrm{ij}}$$
where Y is the response function, A_0_ is a constant, A_i_, A_ii_, and A_ij_ are linear, quadratic, and interaction coefficients, respectively, X_i_ and X_j_ are independent variables, and X_ij_ is interaction coefficients.

For the optimization of encapsulated nisin preparation, a statistical experimental design suggested the preparation of 17 ENs with 12 factorial and five center points (Appendix A).

### 2.4. The Morphology of Encapsulated Nisin and Nanofiber

EN morphology was observed under a SUI510 scanning electron microscope (Hitachi AG, Tokyo, Japan) at 10 kV. The effects of ultrasonication time and nisin mass ratio on the nanofiber morphology were observed under a SUI510 scanning electron microscope (Hitachi AG, Tokyo, Japan) at 10 kV.

### 2.5. Particle Size and Polydispersity Determination

A particle size analyzer (Zetasizer Nano ZS90, Malvern Instruments Ltd., Malvern, UK) was used to measure particle size and polydispersity (PDI). Before the test, the samples were diluted 1:100 with ultrapure water to avoid interparticle interactions and multiple scattering.

### 2.6. Encapsulation Efficiency Determination

The encapsulation efficiencies of Nisin (EE) were determined by an agar diffusion test [9]. The nisin EE was calculated as follows:(2)$$\;\mathrm{EE}\left(\%\right)=\frac{{\mathrm{Nisin}}_{1}}{{\mathrm{Nisin}}_{0}}\times 100$$
where Nisin_0_ is the concentration of nisin in the polymer mixtures., and Nisin_1_ is the concentration of the nisin trapped in the microcapsules.

### 2.7. Turbidity Measurement

EN dispersion served as a turbidity indicator. Absorbance was measured spectrophotometrically at 500 nm and a constant 25 °C.

### 2.8. Nanofiber Preparation

Eight grams PVA powder and 10 g PAAS powder were weighed out. Each was dissolved in 100 mL deionized water and the solutions were mixed. Various amounts of EN were added to the PVA/PAAS mixture at PVA/PAAS powder: EN mass ratios of 100:0, 95:5, 90:10, 85:15, and 80:20. Each solution was stirred for 2 h and ultrasonicated for 0 min, 15 min, and 30 min at 40 kHz and 50 W. The mixture was stirred again for 2 h under the aforementioned conditions to ensure complete reaction.

The solution was then loaded into a metal capillary (0.5 mm i.d.) to prepare fibers through the electrospinning machine (KH-1, Jinan Liangrui Technology Co., Ltd., China). The ejection flow rate was set to 0.5 mL h^−1^ with a syringe pump, the voltage was fixed at 20 kV, and the distance between the capillary tip and the collector was 20 cm. The nanofibers were vacuum-dried at 100 °C for 20 min and conditioned at 25 °C and 50% RH for 24 h before testing.

### 2.9. Fourier-Transform Infrared Experiment

The chemical structures of the nanofibers and the interactions among PVA, PAAS, and EN were examined with a BOEN spectrometer (Feierboen Precision Instruments Ltd., Shanghai, China). The samples were identified at a resolution of 4 cm^−1^, an average of 32 scans, and a range of 4000–650 cm^−1^.

### 2.10. Nanofiber Thickness and Density Measurements

The nanofibers were cut into 10 mm × 30 mm rectangles. Nanofiber thickness (d) was measured at three different points using a gauge with 0.01 mm accuracy. The average of triplicate readings was recorded. Nanofiber mass (m) was determined on an electronic balance with 0.1 mg accuracy. Nanofiber density was calculated as follows:(3)$$\rho =\frac{m}{s\cdot d}$$
where ρ is the density, m is the nanofiber mass, s is the area of nanofiber, and d is the nanofiber thickness.

### 2.11. Determination of Mechanical Properties

The nanofibers were cut into 50 mm × 10 mm strips and their tensile strength (TS) and elongation at break (EAB) were measured with the HD-B609B-S (Haida Instruments Co. Ltd., Guangdong, China). The crosshead speed was 20 mm min^−1^ and the initial distance was 30 mm. All samples were measured in triplicate and the averages were recorded and used in the subsequent analysis.

### 2.12. Swelling Ratio (SR) and Water Vapor Permeability Measurements

The nanofiber samples (20 mm × 20 mm) were cut out, weighed (W_0_), and immersed in distilled water at 25 °C for 24 h. Undissolved material and surface water were removed with filter paper before reweighing (W_1_). The swelling ratio (SR) of nanofiber was calculated as follows:(4)$$\mathrm{SR}=\frac{W_{1}-W_{0}}{W_{0}}$$
where W_1_ is the mass of the swollen sample and W_0_ is the mass of the initial sample.

The wet cup method was adapted to measure nanofiber water vapor permeability (WVP). The samples were secured on cups and placed in a drying tower at 0% RH and 25 °C. The nanofiber WVP was calculated as follows:(5)$$\mathrm{WVP}=\frac{G}{\mathrm{tA}\left(P_{A1}-P_{A2}\right)}$$
where G is the weight loss (g), t is the time (h), A is the nanofiber area (m_2_), and P_A1_ and P_A2_ are the water vapor partial pressures (kPa) inside and outside the cup, respectively.

### 2.13. Surface Color Determination

The nanofiber color characteristics were evaluated with a CS-10 color difference meter (Baiteng Electronic Technology Co. Ltd., Hangzhou, China) using three parallel samples per treatment/condition. The averages of triplicate readings were recorded and used in the subsequent analysis. A standard plate served as the control film (L* = 90.07 ± 0.93; a* = 2.94 ± 0.40; b* = −5.33 ± 0.28). Total nanofiber color difference was calculated as follows:(6)$$\Delta E=\sqrt{{\left(\Delta L^{*}\right)}^{2}+{\left(\Delta a^{*}\right)}^{2}+{\left(\Delta b^{*}\right)}^{2}}$$
where ΔL, Δa, and Δb are the differences between the color values of the standard color plate and the film samples.

### 2.14. Light Transmittance Determination

Nanofiber light transmittance was obtained by testing 10 mm × 50 mm specimens in a UV-visible spectrophotometer (UV-1800; Shimadzu Corp., Kyoto, Japan) at a wavelength range of 200–800 nm. Each specimen was tested in triplicate and the averages were recorded.

### 2.15. Biodegradability Assay

The natural soil buried degradation method of Nguyen et al. was used here [27]. Nanofibers were buried in natural soil and tested weekly. The averages of three samples per group were recorded.

### 2.16. Nisin Release from Nanofibers

Five milligrams of nanofiber containing 15% EN was used in this assay adapted from Bouaziz et al. [28]. At 25 °C, 5 mg of nanofiber was dispersed in 10 mL phosphate-buffered saline (PBS), placed in a dialysis membrane immersed in 200 mL PBS, and continuously stirred at 400 rpm. At predetermined intervals, the PBS was withdrawn, and its absorbance was measured spectrophotometrically. Nisin release from the nanofibers was calculated as follows:(7)$$\mathrm{Release}\left(\%\right)=\frac{\mathrm{Release}\;\mathrm{nisin}}{\mathrm{Total}\;\mathrm{nisin}}\times 100$$

### 2.17. Antimicrobial Experiment

The antibacterial properties of nanofibers were measured using the previous method [29]. The antibacterial activity of nanofiber membrane was evaluated by the agar diffusion method. The target strain was inoculated in the ordinary nutrient liquid culture medium, and was cultured at 37 °C for 24 h, and then continuously diluted to 10^−7^ CFU/ML. Then, 0.10 mL bacterial cells were coated on the surface of tryptone soybean agar (TSA) medium, and three nanofiber discs were placed on the inoculation surface. The plate was cultured at 37 °C for 24 h, and the inhibition zone was measured by digital caliper. The antibacterial test was divided into 3 times and the average value was taken.

### 2.18. Preservation Test

The strawberries from each set of treatment conditions were medium well, and had an intact and fresh appearance. The assessments were conducted under ambient conditions at about 20 ± 1 °C and 60 ± 5% RH in a sensory evaluation room. The changes in the strawberries’ appearance on days 2, 4, 6, and 8 were valued and recorded for qualitative evaluation.

### 2.19. Statistical Analysis

All data are expressed as mean ± standard deviation. Analysis of variance and significance testing were performed using Duncan in statistical analysis software SPSS 22, and mapping was performed with Origin 2017.

## 3. Results

### 3.1. Nisin Encapsulation Optimization via a Response Surface Methodology Experimental Design

The mechanism by which soy oil/gelatin encapsulates nisin is shown in Figure 1. Nisin is water-soluble as its polar groups (-NH_2_, -COOH) form hydrogen bonds with water molecules. Unlike nisin solution, soy oil is nonpolar, has a low surface tension, and creates large interfacial tension between itself and the water phase. Thus, it is insoluble in water. In this experiment, the oil–water interface was visible as a surface rather than a line. The upper layer was oil and the lower layer was water. However, the addition of the lipophilic emulsifier Stepan^®^ 80 reduced the interfacial tension by linking the gap between oil phase and water phase, resulting in free energy decrease for the formation of emulsion, and allowed an interfacial film to form between the oil and water phases. An Oil/Water (O/W) emulsion formed, and nisin solution droplets were trapped in the oil within the bulk oil phase. In this way, a single oil phase and water accumulation were avoided. The resultant Water/Oil (W/O) emulsion droplets were placed in a gelatin solution with the hydrophilic emulsifier Tween^®^ 80 (outer phase). The mixture was ultrasonicated and broken up into smaller droplets because intensive disruptive forces and ultrasound cavitation generated by ultrasonic wave can break up the water and oil phases. Tween^®^ 80 adsorbed at the interface induced attraction and aggregation in response to interfacial disturbances. Its lipophilic group entered the oil phase while its hydrophilic group entered the water phase, Hence, the surface energy increased. In this way, they remained stable in the suspension system for a long time and created a Water/Oil/Water (W/O/W) emulsion.

EN production efficiency can improve with ultrasonication time. Also, pH significantly influences nisin activity. The emulsifier ratio (E_1_/E_2_) also has important effects on EN structure and safety. Tween^®^ 80 and Stepan^®^ 80 were used as emulsifiers here as they have low toxicity, high emulsifiability and dispersibility, and the ability to minimize EN size [30]. For these reasons, the effects of the ultrasonication time, pH, and emulsifier ratio were explored here to create optimal EN. Based on the BBD response surface design, 17 experiments were run with different randomized combinations of the aforementioned variables. The observed and predicted experimental outcomes are listed in Table 1. Multiple regression analysis of the experimental data correlated the response and test variables according to a second-order polynomial equation:(8)$$\mathrm{EE}=+85.41+1.34X_{1}+4.86X_{2}+2.59X_{3}+1.87X_{1}X_{2}-0.65X_{1}X_{3}+1.07X_{2}X_{3}-1.47X_{1}^{2}-9.52X_{2}^{2}-2.10X_{3}^{2}$$

Table 1 shows that this model had highly significant fitness because its F-test had a high F-value (116.14) and a low *p*-value (*p* < 0.0001). The data in Table 1 also suggest that the variables affecting EE were the linear effect of pH, the quadratic effect of ultrasonication time, the quadratic effect of E_1_/E_2_, respectively (*p* ≤ 0.05), and the interaction between E_1_/E_2_ and ultrasonication time (*p* ≤ 0.05). A comparison of the F-values revealed that the order of the factors influencing EN EE was ultrasonication time (279.3) > E_1_/E_2_ (79.21) > pH (21.19). The high values for the determination coefficient (R^2^ = 0.9933) and the adjusted determination coefficient (Adj R^2^ = 0.9848) indicated strong correlation between the study results and the values predicted by this equation. Therefore, RSM models may be used to study the linear, interaction, and quadratic effects of pH, ultrasonication time, and E_1_/E_2_ on nisin EE.

Appendix A shows that the main factor affecting the EE was ultrasonication time. EE initially increased with ultrasonication time but decreased when it was > 3 min. This result corroborates the one reported by Myers et al. [31]. After short ultrasonication times, the nisin solution was uniformly dispersed in the soy oil and could be easily encapsulated. In contrast, excessive ultrasonication damaged the EN, thereby causing the nisin to leak and the EE to decline. When E_1_ > E_2_, the EE increased until it reached a maximum at E_1_/E_2_ = 1.57. The relatively larger quantity of Stepan^®^ 80 caused the nisin to emulsify more effectively in the soy oil, form smaller droplets, and enable greater amounts of nisin to be encapsulated. The results indicated that the optimal conditions for EN preparation were pH = 4.2, ultrasonication time 3 min, and E_1_/E_2_ = 1.57. These settings can maximize nisin EE and optimize EN properties.

### 3.2. Effects of Ultrasonication on Characterization of Encapsulated Nisin 

To examine the effects of ultrasonication on EN characterization, we measured the particle size, PDI, and EE of EN under various ultrasonication times (Table 2). Relative to EN not subjected to ultrasonication, those that were ultrasonicated had significantly smaller particle sizes and PDI. For the untreated EN, diameter = 550 ± 30 nm and PDI = 0.78 ± 0.02. The EN was smallest and had uniform PDI after 3 min of ultrasonication. However, EN particle size increased and PDI decreased with increasing ultrasonication time. Similar results were reported by Tang et al. [32]. As the ultrasonication time rose from 3 min to 5 min, the average particle sizes significantly increased from 320 ± 20 nm to 390 ± 20 nm (*p* < 0.05). The suspended EN particles became unevenly dispersed and their homogeneity decreased with increasing ultrasonication time up to 5 min. Joshi et al. indicated that EN stability and retention time increased with EE [33]. EE increased with ultrasonication time when the time did not exceed 3 min. At 3 min ultrasonication, the maximum EE was 86.66 ± 1.59% possibly because the particle size distribution and the emulsifier adsorption rate on the particle surface are affected by the ultrasonication time of the emulsification process [32]. It is suggested that when the particle size is small, it can possess high entrapment efficiency due to the energy output of ultrasonication. This is probably because intensive disruptive forces and ultrasound cavitation generated by ultrasonic wave can break up the water and oil phases and increase the binding site between oil droplets and the emulsifier, which enhance the oil droplet emulsification and increase the nisin EE [34]. When the ultrasonication time exceeded 3 min, the W/O/W could no longer embedded the excess nisin and the EE consequently decreased to 76.42 ± 1.57%. When the ultrasonication time further increased and surpassed the encapsulation threshold, the nisin either dissolved or was adsorbed to the EN surfaces and the EE declined. It is possible that the sheer force of the ultrasonication to lower the physical properties of the interfacial film and the oil–water interfacial viscosity, also act to destabilize the EN structure. Some researchers reveal that the emulsion particles can be broken in the surpassed ultrasonication process. Our study indicated that a relatively long ultrasonic time was preferable to produce uniform particle sizes and PDI for EN. However, any further increases in ultrasonication time adversely affected EE. Thus, 3 min of ultrasonication was optimal in terms of EE, particle sizes, and PDI.

### 3.3. SEM of Encapsulated Nisin and Nanofiber

The microscopic characterization of EN (pH 4.2, ultrasonication time 3.0 min, and E_1_/E_2_ 1.57) is shown in Figure 2a,b Under optimal conditions, the EN was spherical, smooth, intact, and had sound network structure integrity. The average EN size was 321 ± 20 nm with nisin present. The effects of ultrasonication time on turbidity are shown in Figure 2c. Relative to EN particles that were ultrasonicated, those that were not subjected to ultrasonication had maximum turbidity at 0.72 ± 0.11. EN turbidity was lowest and particle dispersion was widest after 3 min of ultrasonication time. The initial decrease in turbidity was followed by an increase. The suspended EN particles became evenly distributed and their homogeneity increased with ultrasonication time [35]. However, when the ultrasonication time passed a certain point, the homogeneity decreased, possibly because the ultrasonication inhibited EN sedimentation. Joshi et al. reported a similar finding [33].

From left to right, show the nanofiber morphology at ×5000, and ×50,000 (Figure 2d,j). These images reveal that PVA/PAAS/EN-0% nanofibers exhibited good network structural integrity, smoothness, and no cracking or bead defects. However, the nanofibers appeared disorganized after the EN was added. The disordered areas of the PVA/PAAS nanofibers presented with crosslinking. Though the EN-doped nanofibers retained a bead structure and were now wider, their surfaces were smooth and intact. Small bumps were visible in the nanofibers. These may have indicated EN immobilization in the PVA/PAAS. This observation aligned with a previous study which reported that after lysozyme immobilization, bumps were detected on the surfaces of the chitosan nanofibers encapsulating it [36]. The average nanofiber diameters are shown in Figure 2e,g. The diameter of the EN-free PVA/PAAS nanofibers was in the range of 250–350 nm and the average was 290 ± 16 nm. However, when the EN content was increased to 10%, the nanofiber diameter was in the range of 250–450 nm which was greater than that for the PVA/PAAS/EN-0% nanofibers. Further, the average diameter was now 370 ± 23 nm. This particle size increase may be explained by the molecular interactions between PVA and EN and the uneven charge distribution resulting from unstable electrostatic spinning. Khan et al. reported similar findings for PVA/ZnO nanofibers [37]. They found that the nanofiber diameter increased with ZnO content. 

We then conducted a ×50,000 magnification analysis to observe morphological and structural changes after ultrasonication for 0 min, 15 min, and 30 min (Figure 2h–j), and revealed that the compactness of the fiber network structure increased with ultrasonication time. Thus, the treatment increased the number of hydrogen bonds and crosslinking between PVA and EN. For PVA/PAAS nanofibers not subjected to ultrasonication, the inter-fiber pores remained intact and only a few fibers were fractured. After 15 min of ultrasonic treatment, the nanofiber morphology was essentially unchanged but the fiber diameter slightly increased. This observation may be explained by crosslinking after EN incorporation. Hydrogen bond formation occurred among the PVA, PAAS, and EN molecules and improved their TS and EAB value. After 30 min ultrasonication, the nanofibers were disrupted and only a few remained intact and free of pores. Cavitation mainly accounts for this finding which aligns with the report of Qiao et al. [38]. In addition, the observed high specific surface area of the microstructure provides potential for higher bioactivity, thus enabling the obtained product to be used for food preservation [39].

### 3.4. Fourier Transform Infrared Spectroscopy (FTIR)

Figure 3 shows characteristic ATR-FTIR spectra corresponding to nanofibers subjected to different ultrasonication times and EN concentrations. The first peak of nisin was attributed to O-H and N-H axial stretching at 3401 cm^−1^ and the stretching vibration of C-H bond appeared at 2964 cm^−1^. In addition, the weakening of the peak at 1738 cm^−1^ of EN indicated that the combination of soy oil and nisin resulted in the weakening of the characteristic absorption of the C=O group and the NH^3+^ band disappears completely [40]. Figure 3 shows that nisin was embedded in the W/O/W emulsions and was bonded by hydrogen bonding to maintain certain structural characteristics of EN. For the PVA/PAAS/EN nanofibers, the bands at 3370 cm^−1^ and 1650 cm^−1^ were caused by stretching the hydroxyl group and C=C bonds, respectively [41]. The characteristic PVA peaks were at 3292 cm^−1^, 1086 cm^−1^, and 846 cm^−1^. The first peak was attributed to the stretching of the O-H bonds and the last two were the result of a CH_2_ stretching vibration and a peak caused by C-O stretching at 1733 cm^−1^ [42]. Figure 3 shows that the intensities of all absorption peaks increased after ultrasonication because the treatment increased the vibrational dipole moment and the exposed group [43]. The characteristic peaks of the nanofibers after ultrasonication were 3316 cm^−1^ and 3331 cm^−1^. The hydrogen interactions were stronger than those for nanofibers subjected to short-duration- or no ultrasonication. A comparison of the infrared spectra for ultrasonicated and untreated nanofibers indicated that an optimal ultrasonication time strengthened the intensities of the O-H absorption peaks and increased the numbers of hydrogen bonds. This finding was consistent with the observed morphology. As shown in Figure 3, the O-H absorption peaks shifted from 3292 cm^−1^ in the PVA/PAAS/EN-0% nanofibers to 3331 cm^−1^ in the PVA/PAAS/EN-15% nanofibers [44]. Hence, new hydrogen bonds formed and the addition of EN strengthened the intensities of the O-H absorption peaks. This observation corroborates that which was reported by Yuan et al. [45]. However, no significant changes were observed in any other characteristic nanofiber peaks. For this reason, the physical mixing of EN and PVA/PAAS resulted in a complex formation.

### 3.5. Nanofiber Thickness and Density

The PVA/PAAS/EN nanofiber thicknesses are listed in Table 3. The thicknesses were in the range of 0.43 ± 0.02–0.53 ± 0.03 mm which was slightly greater than that determined for the PVA/PAAS nanofibers (0.41 ± 0.01 mm). The EN granules occupy a certain volume and increase the free volume of the PVA/PAAS network, macromolecule mobility, and, by extension, nanofiber thickness. Moreover, the various chemical constituents in EN could widen the gaps between the granules in the substrate, thereby increasing nanofiber thickness. Though the additives tested in previous studies differed from those used here, it is expected that nisin would have a similar impact on nanofiber thickness [46].

The density of PVA/PAAS/EN-0% nanofibers without ultrasonication was 0.37 ± 0.03 g cm^−3^. EN addition produced PVA/PAAS/EN-15% nanofibers whose untreated density was 0.60 ± 0.05 g cm^−3^. Nanofiber density significantly (*p* < 0.05) increased with EN content (Table 3). This discovery matched the one reported by Jafarzadeh et al. [47]. The addition of EN to the PVA/PAAS network might have increased the solid content. The PVA/PAAS/EN-0% nanofibers subjected to < 30 min of ultrasonication had a density of only 0.35 ± 0.03 g cm^−3^. At 15% EN and 15 min of ultrasonication, the nanofiber density increased to 0.77 ± 0.04 g cm^−3^ which was ~2× improvement over that of the untreated PVA/PAAS/EN-0% nanofibers. After 15 min of ultrasonication, the nanofiber density significantly increased (*p* < 0.05). On the other hand, after 30 min of ultrasonication, the nanofiber density declined possibly because prolonged ultrasonication induced cavitation. After brief ultrasonication, the nanofiber network structure condensed and became inseparable because of the pressure created by ultrasonication and the EN dispersal [48]. After long ultrasonication (30 min), the molecular agitation was increased, the nanofiber bonds were broken, the network structure loosened, and the nanofibers underwent cavitation [49]. The net effect was a decrease in nanofiber density. However, increasing the EN content from 5% to 20% did not increase nanofiber density.

### 3.6. Mechanical Properties of Nanofibers

The previous experiment revealed that the mechanical properties of the nanofibers were changed by ultrasonication and the applicability of untreated nanofibers is limited [50]. The influence of various nisin encapsulation levels on PVA/PAAS nanofiber TS and EAB are listed in Table 3. EN addition significantly altered the nanofiber mechanical properties (*p* < 0.05). The TS values increased with EN content. Ebrahimnezhad-Khaljiri et al. reported similar findings [51]. The TS of the untreated PVA/PAAS/EN-15% nanofibers was 10.35 ± 0.42 MPa and was higher than that for the untreated PVA/PAAS/EN-0% nanofibers (7.09 ± 0.33 MPa). On the one hand, the increased EN content caused strong hydrogen interactions between the functional groups in the EN and the PVA and PAAS hydroxyl groups. FTIR disclosed a compact structure and gradual improvement in PVA/PAAS nanofiber TS with increasing EN. At very high EN levels, though, the EN aggregated, reduced the surface free energy, weakened certain parts of the nanofibers, decreased hydrogen bonding among EN, PVA, and PAAS, and reduced nanofiber TS [52]. Table 3 shows that the PVA/PAAS/EN-0% nanofiber EAB was 69.31 ± 2.91%. The PVA/PAAS nanofiber EAB increased with EN content, reached a maximum of 130.08 ± 3.63% at 15% EN, and visibly decreased thereafter. The highest EN concentrations may have created strong hydrogen interactions, enhanced aggregation, impeded chain movement, and reduced plasticizing and nanofiber flexibility. These effects were evident from the nanofiber morphology (Figure 2). The foregoing observations indicate that EN addition improved the mechanical properties (especially TS and EAB) of the nanofibers. Similar results were reported by Cano et al. [53].

Ultrasonication also affected the mechanical properties of the nanofibers. After 15 min of ultrasonication, the nanofiber TS and EBA significantly increased (*p* < 0.05). After 30 min of ultrasonication, however, the mechanical properties of the nanofibers deteriorated. A previous study indicated that the material could become inseparable in response to prolonged ultrasonication [54]. Therefore, EN dispersal in the PVA/PAAS nanofiber and strong hydrogen interactions between the EN and the nanofiber could account for the measured increases in TS and EAB. In fact, the nanofiber TS and EAB declined because the extended ultrasonication increased molecular motion and broke the hydrogen bonds [55]. Ultrasonication also smoother the nanofiber surface (Figure 2), increased adhesion between the nanofibers, and improved their mechanical properties. This conclusion was in agreement with the one reported by Chen et al. [56].

### 3.7. Swelling Ratio

Due to the high SR of PVA and PAAS, various mechanical properties of the nanofibers made from them are easily changed by environmental humidity. Here, we tried to optimize the swelling ratio by ultrasonication (Table 3). The SR of the PVA/PAAS/EN nanofibers was in the range of 221.65 ± 3.57–335.35 ± 4.84%. Untreated PVA/PPAS nanofibers immediately swelled in the presence of water. However, nanofiber SR decreased with increasing EN. It might be due to certain PVA hydroxyl groups in the nanofibers formed hydrogen bonds with the EN, reduced the number of strong hydrophilic hydroxyl groups, lowered the relative hydrophilicity in the nanofibers, and reduced their SR. Ultrasonication significantly accelerated the motion of the PVA chain, induced crystallization, and altered the nanofiber SR [16]. The SR of the nanofibers subjected to 30 min of ultrasonication was lower than that for those exposed to 15 min of ultrasonication. The observed decrease in nanofiber SR in ultrasonication was primarily the result of changes in the hydrogen bond and hydrophilic hydroxyl group content.

### 3.8. Water Vapor Permeability

Table 3 shows that the nanofiber WVP initially decreased and then increased with increasing EN. When the EN was added to the nanofibers, the WVP after 0 min or 15 min of ultrasonication were significantly lower than that for the PVA/PAAS/EN-0% nanofibers (*p* < 0.05). When the EN content was increased to 15%, the WVP fell to a minimum of 1.02 × 10^−3^ ± 0.03 (g·h^−1^ m^−2^ pa^−1^). This value was 28.67% lower than that for the PVA/PAAS/EN-0% nanofibers. The original PVA and PAAS polymer structures may have changed after the EN and nanofibers were combined because the EN might have extended the path through which the water molecules had to pass [57]. However, this result differed from the effect of EN on film WVP reported by Alves possibly because the network structures of nanofibers and films are not the same [58]. Nevertheless, the nanofiber WVP increased to 1.25 × 10^−3^ ± 0.03 (g·h^−1^ m^−2^ pa^−1^) after 20% EN was added. The addition of very large proportions of EN could increase WVP by modifying the PVA/PAAS network structure and facilitating moisture permeation inside and outside the pores [59]. Ultrasonication also changed the nanofiber WVP. After 15 min ultrasonication, the connections among EN, PVA, and PAAS increased, the nanofibers became more compact and the WVP decreased. This observation was supported by the density measurements. This effect on nanofiber WVP was augmented further still after 30 min ultrasonication and reached a maximum of 1.84 × 10^−3^ ± 0.04 (g·h^−1^ m^−2^ pa^−1^). Prolonged ultrasonication may have disrupted most of the nanofibers, leaving only a few intact and without pores (Figure 2). In this way, the nanofiber pores expanded and facilitated moisture permeation [60]. WVP is a crucial feature in the selection of packaging materials to control the water transfer between the environment and food. Generally, packaging materials with low WVP values are preferred for packaging food. Low WVP value provides high barrier for the application of PVA/PAAS/EN nanofibers in food packaging [61].

### 3.9. Nanofiber Color

Nanofiber color is important as it directly influences consumer acceptability and packaged product appearance. Based on the standard white plate (L* = 90.07 ± 0.93, a* = 2.94 ± 0.40, b* = −5.33 ± 0.28), the effects of EN content and ultrasonication on nanofiber color difference (ΔE) are presented in Table 4. L*, a*, and b* for the PVA/PAAS/EN-0% nanofiber was 95.37 ± 0.22, 4.00 ± 0.64, and −1.77 ± 0.05, respectively. However, Xie et al. reported 92.59 ± 0.04, 0.51 ± 0.04, and −6.05 ± 0.04 [53]. Hence, nanofibers were more brightly colored than casting films. L* and a* were lower for PVA/PAAS/EN-5% than PVA/PAAS/EN-0%. Therefore, the nanofibers became darker after EN was incorporated into them. Nanofiber blueness increased with EN content. For the PVA/PAAS/EN-0% nanofibers, b* was −1.77 ± 0.05 whereas for the PVA/PAAS with EN it was −3.01 ± 0.24. The ΔE, a*, and b* all declined with increasing EN content. Thus, EN addition enhanced the blue–green coloration in the nanofibers [62]. Ultrasonication also affected nanofiber color. Ultrasonication decreased nanofiber L*, a*, b*, and ΔE relative to untreated nanofibers (Table 4). The ultrasonicated nanofibers tended towards a brownish color. An optimal ultrasonication time could result in a uniform and compact nanofiber structure by dispersing the EN and altering the nanofiber color [63]. However, excessive ultrasonication could further deepen the nanofiber color by causing the EN to aggregate [25].

### 3.10. Light Transmittance

Arfat et al. measured nanofiber light transmittance at 600 nm [64]. Nanofiber light transmittances measured here are listed in Table 4. The PVA/PAAS/EN-0% nanofibers had a strong light barrier and their transmittance was 9.18 ± 0.44%. Under normal conditions, the crystalline and amorphous regions coexist in PVA/PAAS nanofibers. Light cannot directly pass through a crystalline polymer, so refraction and reflection occurred at the PVA/PAAS nanofiber interface [55]. Further, nanofibers fabricated by electrospinning were opaque [65]. Light transmittance decreased with increasing EN content. Rouhi et al. reported similar results for ZnO-incorporated gelatin films [66]. Nanofiber granules such as the EN in the interstitial spaces between fibers could hinder light transmittance. Table 4 shows that transmittances significantly decreased with increasing EN content (*p* < 0.05), reaching a minimum of 7.76 ± 0.12% at 15% EN but increasing to 8.38 ± 0.27% at 20% EN. Thus, the spatial structure of the PVA/PAAS/EN nanofibers was altered by very high EN concentrations. The observed reduction in light transmittance may have been the result of refraction and reflection from agglomerated EN particles embedded in the nanofibers. Ultrasonication also affected nanofiber light transmittance. After 15 min of ultrasonication, transmittance significantly decreased. EN distribution in the nanofibers became more uniform after ultrasonication and prevented light from passing through. However, when the ultrasonication time was increased to 30 min, transmittance also increased. Moreover, extended ultrasonication disrupted the nanofiber structure and facilitated light transmittance [67].

### 3.11. Biodegradability

The soil burial degradation assay was used to evaluate PVA/PAAS/EN nanofiber biodegradability in natural environments. In general, humidity and the chemical structure of materials influence their biodegradability [68]. Therefore, it was expected that both EN content and ultrasonication time would affect biodegradability. Figure 4 shows that EN-doped nanofiber degradation was slower than that of EN-free nanofibers. Degradation gradually increased initially and then rapidly increased after 2 weeks. After 4 weeks, the degradation rates of the PVA/PAAS/EN-5% and PVA/PAAS/EN-20% nanofibers were 20.94 ± 0.43% and 19.81% ± 0.42%, respectively. These rates corresponded to 8.87% and 13.79% reductions relative to PVA/PAAS/EN-0% nanofibers (22.98 ± 0.38%). These findings concurred with those of other studies [69]. After 4 weeks soil burial, the PVA/PAAS/EN-0% nanofibers presented with the highest weight loss rate possibly because of PVA and PAAS hydrophilicity (Table 3). A large amount of soil moisture entered the fibrous structure, altered the nanofiber network, and facilitated degradation by soil microorganisms [70]. In contrast, the nanofiber degradation rates significantly decreased with increasing EN content. The interaction between PVA/PAAS and EN reduced the relative hydrophilicity of the nanofiber constituents, the overall nanofiber SR (Table 3), and the biodegradation rate. However, antimicrobial activity gradually increased with nanofiber EN content, reduced soil microorganism activity, and attenuated nanofiber dissolution. Figure 4 shows that the nanofiber degradation rate changed in response to ultrasonication. Relative to untreated nanofibers, those subjected to 15 min ultrasonication had lower degradation rates. The degradation rate for the nanofibers containing 5% EN was only 18.94% ± 0.39% which was 9.55% lower than that of the untreated nanofibers. After 30 min of ultrasonication, the PVA/PAAS/EN nanofiber degradation rates were higher than those of the untreated nanofibers. This discovery resembled that which was reported by Liu et al. [25]. Ultrasonication may alter the nanofiber swelling ratio, structure, and antimicrobial activity which, in turn, could affect the nanofiber degradation rate.

### 3.12. Nisin Release from Nanofibers

Nisin release mainly determines nanofiber antibacterial activity. We compared the rates of nisin release from PVA/PAAS/EN-15% nanofibers under various ultrasonication times. Nisin was released from the nanofiber by surface erosion, decomposition, diffusion, and desorption [71]. Figure 5a shows the amount of nisin released from the nanofiber into PBS media over 15 d. All samples presented with sustained nisin release for the first week. This observation was consistent with that reported by Monjazeb-Marvdashti et al. [72]. The initial quantity of nisin released from nanofibers was large. The nisin freely dispersed in the PBS and demonstrated a stable and continuous release pattern over 7 d. The initially higher relative nisin release rate may be explained by the fact that the nisin was attached to the outer edges of the EN and nanofiber [73]. The subsequent decline in the nisin release rate could be attributed to the fact that the nisin diffused from the inner core of the EN. Ultrasonication may alter physicochemical and biological properties. Therefore, the ultrasonication time was expected to influence nisin release. The nisin release rate was lowest in the absence of ultrasonication. As ultrasonication time increased from 0 min to 15 min, the nisin release rate significantly increased from 74.38 ± 2.66% to 85.28 ± 2.38% (*p* < 0.05). In contrast, when the ultrasonication exceeded 30 min, the nisin release rate declined to 80.19 ± 2.28%. Prolongation of ultrasonication time increases nanofiber SR, which could enhance the dissolution of the EN surfaces and improve nisin release.

The release behavior of nisin from the nanofibers was analyzed using three common kinetic model, including first order models, Higuchi, and Korsemeyer–Peppas (KP) models (Table 5). Figure 6a shows the actual release behavior of nisin from nanofibers tracked spectrophotometrically. Obviously, nisin was almost completely released after a release period of 16 days. Considering this release behavior, first order kinetic, a typical model based on the relationship between the release rate and the concentration of the substances participating in or related to the reaction, was firstly applied to predict the release kinetics. According to the correlation coefficient (>0.90) in Table 5 and the fitting curve in Figure 6b, it indicated that the first order kinetic model can better simulate the release behavior of nisin. However, comparing the correlation coefficient (<0.90) in Table 5 and the fitting curve in Figure 6c, it indicates that the release of nisin did not follow the Higuchi model, a typical model based on Fick’s laws of diffusion through a planar system. Furthermore, we found that the fitting curve only fits well during the rapid release phase, which may be due to the interfacial film of the EN, and the complex spatial structure of the nanofibers, which leads to the release of nisin, did not follow the classical release model in a planar system. Therefore, the release may occur in a three-dimensional system. Based on this hypothesis, the KP model was selected to further fit the release of nisin. Obviously, the fitted curve was consistent with the actual release one (Figure 6d) during the release of nisin, and excellent correlation coefficients were calculated as 0.98814. This may be mainly due to the fact that when the nanofiber was contacted with moisture, water molecules would expand the surface mesh of the nanofibers and enter the nanofiber through the surface mesh, then dissociate sodium carboxylate of PAAS into -COO^−^ and Na^+^, which accelerated the rate that the nanofibers absorbed water due to high osmotic pressure (Figure 5b). Subsequently, a small part of water molecules entering the nanofiber were bound to polymer molecules, most of them existed in the form of free water and were in touch with the EN embedded in the nanofiber. Although there was an oil–water interface phase on the surface of EN, water molecules could pass and change the phase, which caused the EN became thinner and crispier. On the other hand, Na^+^ ions could weak the electrostatic interaction between the EN the water phase and oil phase. Gelatin become soluble in the free water absorbed in the nanofiber and leave the interface due to the electrostatic interaction, thereby lowering its interfacial viscosity and the rigidity of interfacial film. At the same time, it is possible for the water molecules to penetrate the oil film channels, lead to film swelling, thin out, and even break the interfacial film. After that, emulsion droplet coalescence can reduce the interfacial area of emulsion phase and aggravate the emulsion kinetics instability, resulting in destruction of emulsion and EN structure. Consequently, nisin slowly diffuses from the surface mesh into the water.

### 3.13. Antimicrobial Activity

The PVA/PAAS/EN-15% nanofibers had optimal mechanical, light transmittance, and barrier properties. Hence, PVA/PAAS/EN-15% was selected for the antimicrobial efficacy assay using TSA culture medium and stored at 4 °C for 16 days. The antibacterial activity of untreated and ultrasonicated PVA/PAAS/EN-15% nanofibers on *E. coli* and *S. aureus* are shown in Figure 7. EN-free nanofibers did not perceptibly inhibit *E. coli* or *S. aureus.* The incorporation of EN into the nanofibers substantially increased their antibacterial activity. As expected, the PVA/PAAS/EN-15% had lower activity against *E. coli.* than *S. aureus.* Premanathan et al. reported similar observations [74]. The main antibacterial constituent in the nanofiber was the EN and the nanofibers had greater inhibitory efficacy against *S. aureus* than *E. coli*. These conclusions align with those reported in other studies [75]. The antimicrobial activity of the nanofibers doped with nisin by electrospinning provided a 99.99% *S. aureus* kill rate relative to other membranes without nisin [4]. The EN may have attenuated cell permeability by membrane lipid and protein agglutination. In this way, the EN disrupted the bacterial cell membranes, perturbed their metabolism, caused their cell contents to leak, inhibited cell growth, and induced cell death [76]. The EN displayed stronger bacteriostatic and antibacterial activity against Gram-positive *S. aureus* than Gram-negative *E. coli.* The cell wall of the latter bacterium is protected by a lipopolysaccharide (LPS) layer. The nisin constituent in EN has broad-spectrum antibacterial properties. This property was previously reported by Hassan et al. [9]. Nanofibers under different ultrasound times showed significant antibacterial activity (*p* < 0.05). The population of *E. coli* and *S. aureus* decreased from 6.8 ± 0.25 and 7.2 ± 0.31 to 5.29 ± 0.24 and 4.69 ± 0.27 log CFU/g respectively at the end of storage period used to assess antimicrobial activity. The bacterial inhibition rate varied considerably with nanofiber ultrasonication time. Nevertheless, prolonged ultrasonication reduced relative nanofiber antibacterial activity. Ultrasonication may alter nanofiber network structures and the nisin release rate. The latter was confirmed by the nisin release measurements [49]. The research on nisin based fiber membranes is less, and its application in the food industry is relegated to cheese packaging and meat preservation. The research of Meral et al. shows that nano mat containing nisin and curcumin can extend the shelf life of fish fillets to 12 days, effectively improving their quality [77]. The nisin containing nanofibers prepared by Soto et al. [21] and Cui et al. [22]. can effectively inhibit the microbial activity on the surface of cheese and prolong the shelf life of cheese. Moreover, the unique nanostructure of the nanofibers helps to release nisin. Therefore, the antibacterial nanofibers are expected to be used as food preservative packaging. PVA/PAAS/EN can still release strong antibacterial activity in 16 days. Wrapping the food surface with PVA/PAAS/EN fibers can provide long-term antibacterial protection and effectively extend the shelf life of food. The super water absorption of PAAS helps to keep the food surface dry and trigger the release of nisin, which has a broad development prospect.

### 3.14. Preservation Test

To investigate the application potential of PVA/PAAS/EN nanofibers in food preservation, the prepared nanofibers were utilized to pack strawberries. Figure 8 presents the appearance of the strawberries during the 8 days of storage. Strawberry quality decreased with time increasing in all groups. According to the results, strawberry quality decreased as the time increased in all groups; the control and PE-packaged strawberries both had an unacceptable appearance after two days and four days of storage respectively, while the PVA/PAAS/EN nanofibers maintained excellent surface. This means that the nanofibers can effectively extend the shelf life of strawberries for 6 days, which is significant for strawberry deterioration. It is interesting that the PE-packed strawberries began to decay after 4 days, and the decay rate was faster than that of the control. As shown in Figure 8, the surface of the PE-packed strawberry was completely rotten and covered with mold, which is because PE acted as a semipermeable barrier against gas and water and accelerated microbial growth. This effect is consistent with the study by Jiang et al. [78]. However, PVA/PAAS/EN nanofibers are so soft and absorbent that they not only absorb the water from the surface of strawberries effectively but also inhibit microbial growth due to the nisin release.

## 4. Conclusions

Here, RSM was used to optimize EN with maximum encapsulation efficiency. It was determined that the conditions required to achieve this goal were E_1_/E_2_ = 1.57, pH = 4.2, and ultrasonication time = 3 min. The product had remarkably high EE (86.66 ± 1.59%), was small in size (320 ± 20 nm), and had a low polydispersity index (0.27) based on the subsequent examination of the effects of ultrasonication on the EN properties. We prepared PVA/PAAS/EN nanofibers by electrospinning and elucidated the effects of ultrasonication and EN incorporation on their structural and physical properties and antimicrobial activity. In the presence of EN, the nanofiber antimicrobial activity was enhanced, the network structure was more compact, and the morphology was improved according to the scanning electron microscopy (SEM) inspection of the ultrasonication treatment. Moreover, swelling was reduced because the hydrogen interactions between the EN and PVA/PAAS and crystallinity increased as a result of the ultrasonication. EN also doubled the tensile strength of the PVA/PAAS/EN-15% nanofibers (10.35 ± 0.32 MPa) relative to that of the PVA/PAAS nanofibers and increased the nanofiber density to a maximum of 0.77 ± 0.04 g cm^−3^. The release behavior of nisin from EN embedded in nanofibers fit the Korsemeyer–Peppas (KP) model; the maximum nisin release rate of 85.28 ± 2.38% was achieved over 16 days. In this way, nanofibers showed robust antimicrobial activity against *E. coli* and *S. aureus*. The best results were obtained with PVA/PAAS/EN-15% nanofibers. The fiber can effectively inhibit the activity of food microorganisms and provide research direction for new antibacterial food packaging materials.

## Figures and Tables

**Figure 1 nanomaterials-10-01803-f001:**
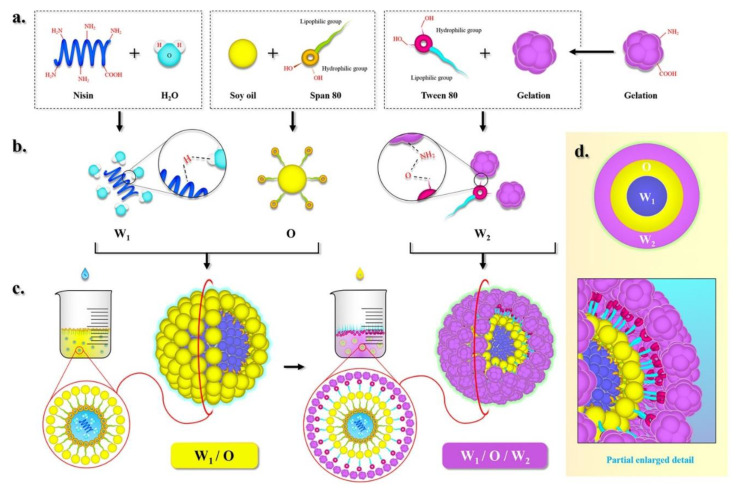
The mechanism by which soy oil/gelatin encapsulates nisin, (**a**) dissolve; (**b**) microscopic reaction; (**c**) macroscopic reaction; (**d**) particle enlarged detail.

**Figure 2 nanomaterials-10-01803-f002:**
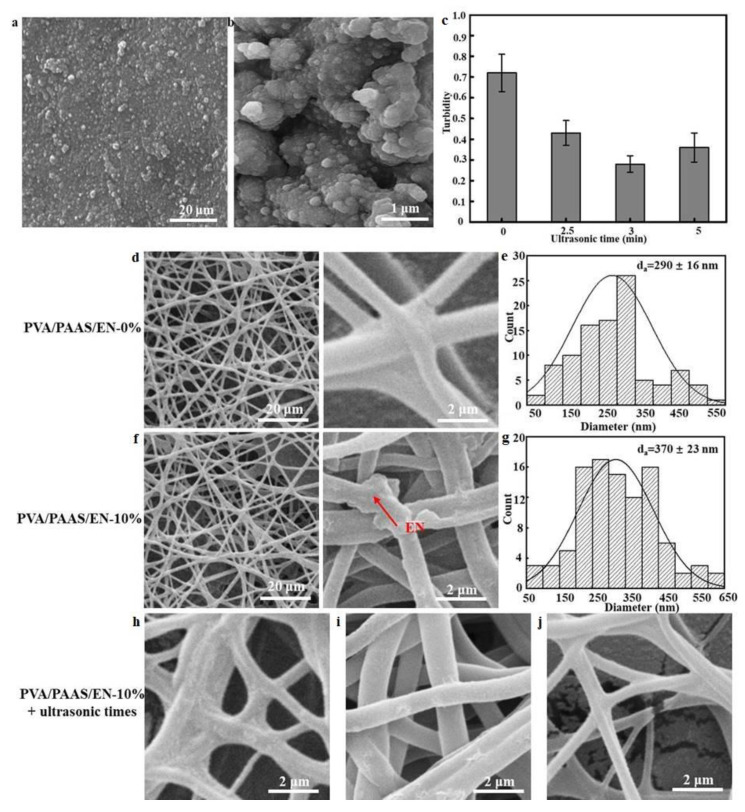
(**a**) SEM image of encapsulated nisin (pH 4.2, ultrasonication time 3.0 min, and E_1_/E_2_ 1.57), (**b**) magnified images of encapsulated nisin; (**c**) The effects of ultrasonication time on turbidity; (**d**) Typical SEM images of PVA/PAAS/EN-0%; (**e**) nanofiber diameter distribution histogram of PVA/PAAS/EN-0%;(**f**) SEM images of PVA/PAAS/EN-10%; (**g**) nanofiber diameter distribution histogram of PVA/PAAS/EN-10%; (**h**) typical SEM 50000x images of PVA/PAAS/EN-10% nanofibers under ultrasound time 0 min; (**i**) ultrasound time: 15 min; (**j**) ultrasound time: 30 min.

**Figure 3 nanomaterials-10-01803-f003:**
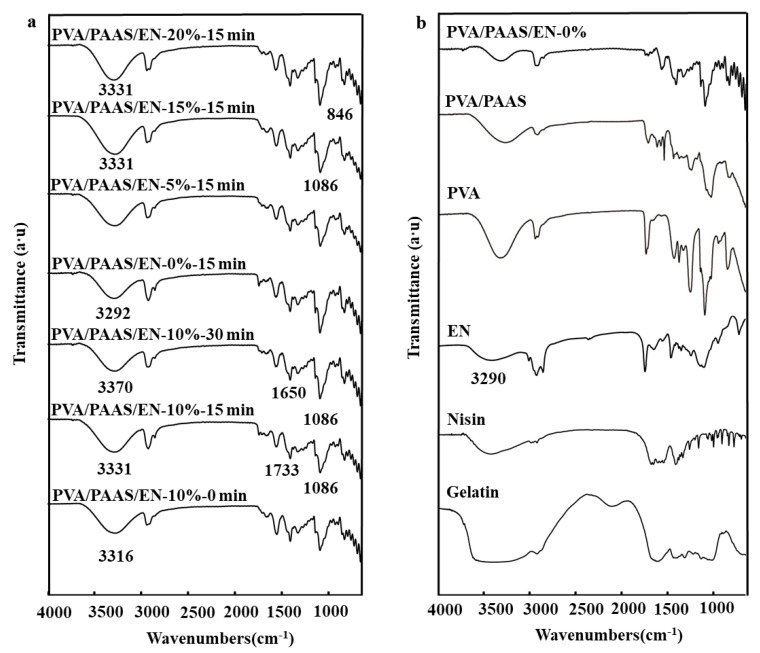
(**a**) FTIR spectra of typical PVA/PAAS/EN-10% nanofibers under different ultrasound times, and PVA/PAAS/EN nanofibers with different EN concentration (under 15 min of ultrasonic treatment); (**b**) FTIR spectrum of PVA/PAAS/EN-10%, PVA/PAAS, PVA, EN, Nisin, Gelatin.

**Figure 4 nanomaterials-10-01803-f004:**
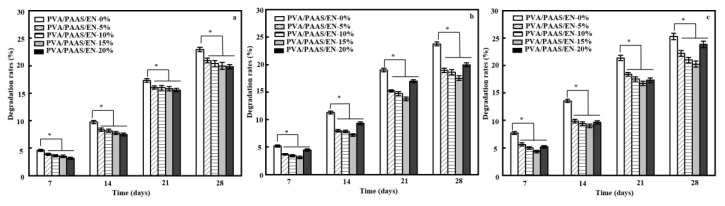
Degradation rates of nanofibers under different ultrasound times: (**a**) 0 min, (**b**) 15 min, and (**c**) 30 min. (* *p* < 0.05, *n* = 5).

**Figure 5 nanomaterials-10-01803-f005:**
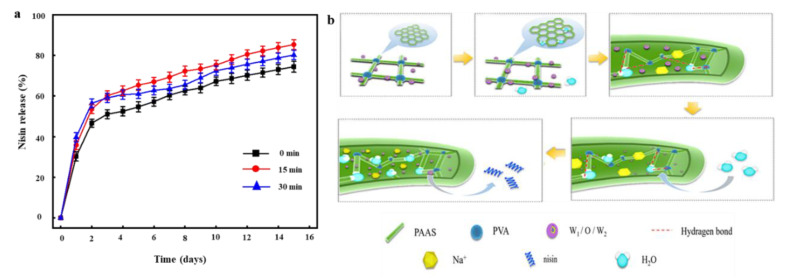
(**a**) The release of nisin from EN and PVA/PAAS/EN-15% nanofibers under different ultrasound times; (**b**) the release behavior of nisin from polyvinyl alcohol/polyacrylate sodium nanofibers.

**Figure 6 nanomaterials-10-01803-f006:**
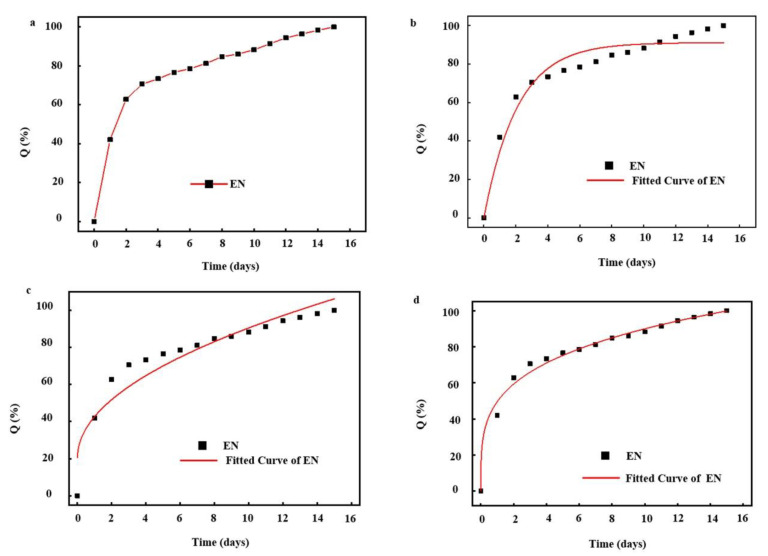
(**a**) Actual kinetic release profiles of nisin from nanofibers; the corresponding fitted curves following first order (**b**), Higuchi (**c**), and Korsemeyer–Peppas (**d**) models.

**Figure 7 nanomaterials-10-01803-f007:**
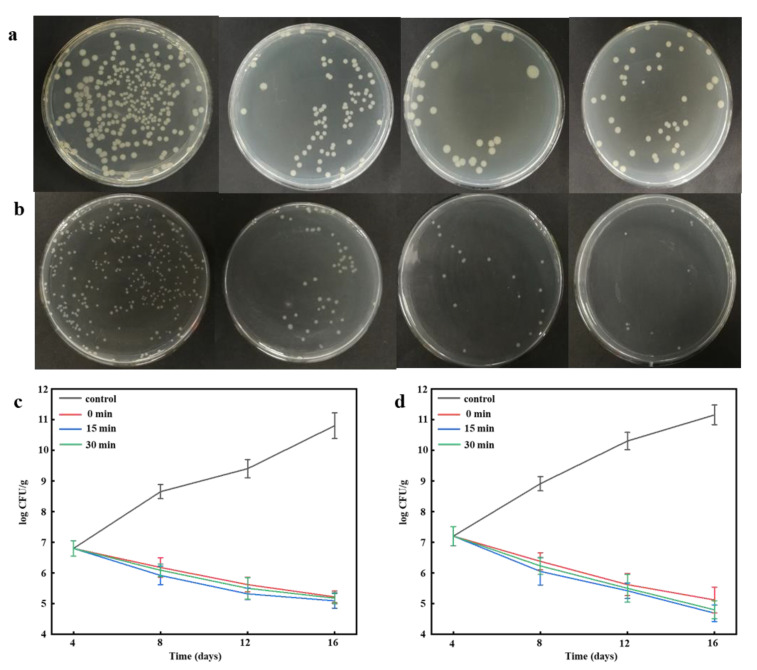
Effect of nanofibers on the growth of (**a**) *E. coli* and (**b**) *S. aureus* on TSA medium stored at 4 °C; Survival of (**c**) *E. coli* and (**d**) *S. aureus* in nanofibers under different ultrasound times throughout storage time at 4 °C for 16 days.

**Figure 8 nanomaterials-10-01803-f008:**
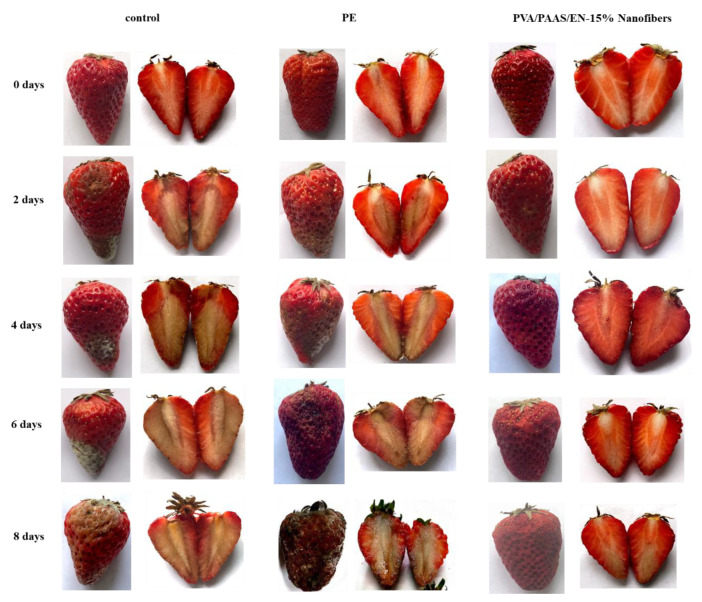
Digital images of the appearance and inner changes of strawberries during storage times.

**Table 1 nanomaterials-10-01803-t001:** Analysis of Variance (ANOVA) for the response surface model.

Source	Sum of Squares	df	Mean Square	F Value	*p*-Value Prob > F
Model	706.07	9	78.45	116.14	<0.0001 ^a^
X_1_	14.31	1	14.31	21.19	0.0025 ^b^
X_2_	188.67	1	188.67	279.30	<0.0001 ^a^
X_3_	53.51	1	53.51	79.21	<0.0001 ^a^
X_1_X_2_	14.06	1	14.06	20.82	0.0026 ^b^
X_1_X_3_	1.69	1	1.69	2.50	0.1577 ^c^
X_2_X_3_	4.56	1	4.56	6.75	0.0355 ^b^
X_1_^2^	9.05	1	9.05	13.39	0.0081 ^b^
X_2_^2^	381.86	1	381.86	565.30	<0.0001 ^a^
X_3_^2^	18.54	1	18.54	27.44	0.0012 ^b^
Residual	4.73	7	0.68	-	-
Lack of Fit	2.22	3	0.74	1.18	0.4210c
Pure Error	2.50	4	0.63	-	-
Cor Total	710.80	16	-	-	-
R^2^ = 0.9961	-	-	-	-	-
Adj R2 = 0.9912	-	-	-	-	-

X_1_ = coded value of pH, X_2_ = coded value of ultrasonic time, and X_3_ = coded value of E_1_/E_2_ (min), ^a^ very significant *p* < 0.01, ^b^ significant *p* < 0.05 and ^c^ not significant *p* > 0.05.

**Table 2 nanomaterials-10-01803-t002:** Particle sizes, encapsulation efficiency (EE), and polydispersity (PDI) values of encapsulated nisin (EN) under different ultrasound times.

Samples	pH	UT (min)	E_1_/E_2_	EE (%)	Particle Sizes (nm)	PDI
1	4.2	0	1.57	72.19 ± 1.82 ^c^	550 ± 30 ^a^	0.78 ± 0.02 ^a^
2	4.2	2.5	1.57	85.34 ± 1.62 ^a^	450 ± 30 ^b^	0.42 ± 0.02 ^b^
3	4.2	3.0	1.57	86.66 ± 1.59 ^a^	320 ± 20 ^d^	0.27 ± 0.01 ^d^
4	4.2	5.0	1.57	76.42 ± 1.57 ^b^	390 ± 20 ^c^	0.36 ± 0.01 ^c^

Different letters in the same column indicate significant differences (*p* < 0.05).

**Table 3 nanomaterials-10-01803-t003:** Thickness, density, mechanical properties, swelling ratio (SR), and water vapor permeability (WVP) values of nanofibers under different ultrasound times.

Samples	Ultrasonic Time (min)	PVA/PAAS/EN-0%	PVA/PAAS/EN-5%	PVA/PAAS/EN-10%	PVA/PAAS/EN-15%	PVA/PAAS/EN-20%
Thickness (mm)	0	0.41 ± 0.01 ^d^	0.43 ± 0.02 ^d^	0.49 ± 0.01 ^c^	0.57 ± 0.01 ^a^	0.53 ± 0.02 ^b^
	15	0.47 ± 0.02 ^c^	0.47 ± 0.02 ^c^	0.56 ± 0.02 ^b^	0.64 ± 0.01 ^a^	0.61 ± 0.03 ^a^
	30	0.44 ± 0.02 ^a^	0.56 ± 0.03 ^bc^	0.52 ± −0.01 ^b^	0.59 ± 0.02 ^a^	0.55 ± 0.03 ^bc^
Density (g/cm^3^)	0	0.37 ± 0.03 ^c^	0.45 ± 0.03 ^b^	0.49 ± 0.02 ^b^	0.60 ± 0.05 ^a^	0.51 ± 0.04 ^b^
	15	0.42 ± 0.04 ^d^	0.58 ± 0.02 ^c^	0.69 ± 0.02 ^b^	0.77 ± 0.04 ^a^	0.67 ± 0.04 ^b^
	30	0.35 ± 0.03 ^c^	0.55 ± 0.04 ^b^	0.57 ± 0.03 ^ab^	0.62 ± 0.02 ^a^	0.59 ± 0.03 ^ab^
TS (MPa)	0	5.81 ± 0.13 ^e^	8.52 ± 0.48 ^b^	7.91 ± 0.39 ^c^	10.35 ± 0.32 ^a^	6.44 ± 0.09 ^d^
	15	6.58 ± 0.27 ^e^	9.21 ± 0.27 ^c^	10.58 ± 0.24 ^b^	12.12 ± 0.49 ^a^	7.76 ± 0.26 ^d^
	30	4.76 ± 0.26 ^d^	6.00 ± 0.26 ^c^	8.33 ± 0.41 ^b^	9.36 ± 0.28 ^a^	4.87 ± 0.33 ^d^
EAB (%)	0	69.31 ± 2.91 ^d^	84.79 ± 3.62 ^c^	92.67 ± 3.54 ^b^	105.69 ± 2.91 ^a^	94.56 ± 2.83 ^b^
	15	77.30 ± 2.59 ^a^	115.75 ± 3.71 ^b^	101.41 ± 3.83 ^c^	130.08 ± 3.63 ^a^	120.69 ± 3.68 ^b^
	30	70.14 ± 2.88 ^d^	98.65 ± 2.45 ^c^	110.94 ± 3.15 ^b^	125.27 ± 2.97 ^a^	114.36 ± 3.72 ^b^
SR (%)	0	324.45 ± 10.25 ^a^	302.26 ± 9.38 ^b^	285.11 ± 11.07 ^bc^	266.53 ± 10.68 ^c^	243.91 ± 11.04 ^d^
	15	335.35 ± 9.51 ^a^	305.47 ± 10.21 ^b^	289.91 ± 12.07 ^bc^	271.82 ± 9.99 ^cd^	256.96 ± 8.65 ^d^
	30	301.59 ± 10.32 ^a^	279.36 ± 10.97 ^b^	255.39 ± 10.25 ^c^	248.16 ± 8.41 ^c^	221.65 ± 9.57 ^d^
WVP × 10^−3^ (g/h m^2^ Pa)	0	1.43 ± 0.08 ^a^	1.24 ± 0.04 ^b^	1.19 ± 0.03 ^b^	1.07 ± 0.03 ^c^	1.25 ± 0.03 ^b^
	15	1.39 ± 0.04 ^a^	1.17 ± 0.07 ^b^	1.03 ± 0.02 ^c^	1.02 ± 0.02 ^c^	1.23 ± 0.03 ^b^
	30	1.57 ± 0.05 ^b^	1.43 ± 0.03 ^c^	1.25 ± 0.03 ^a^	1.33 ± 0.03 ^d^	1.84 ± 0.04 ^a^

Different letters in the same column indicate significant differences (*p* < 0.05).

**Table 4 nanomaterials-10-01803-t004:** Color and light transmittance values of nanofibers under different ultrasound times.

Sample	Ultrasonic Time (min)	PVA/PAAS/EN-0%	PVA/PAAS/EN-5%	PVA/PAAS/EN-10%	PVA/PAAS/EN-15%	PVA/PAAS/EN-20%
L	0	95.37 ± 0.22 ^a^	92.90 ± 1.02 ^ab^	92.04 ± 2.20 ^b^	92.02 ± 1.09 ^b^	87.24 ± 1.73 ^c^
	15	84.87 ± 1.38 ^c^	92.44 ± 2.25 ^a^	80.86 ± 0.47 ^d^	89.42 ± 1.54 ^b^	85.58 ± 0.78 ^c^
	30	85.83 ± 2.26 ^b^	83.31 ± 1.24 ^c^	88.81 ± 0.57 ^a^	87.22 ± 1.52 ^ab^	85.17 ± 2.66 ^b^
a	0	4.00 ± 0.64 ^a^	1.08 ± 0.11 ^b^	0.27 ± 0.07 ^bc^	0.78 ± 0.22 ^bc^	−0.59 ± 0.18 ^c^
	15	−0.46 ± 0.12 ^ab^	1.66 ± 0.23 ^a^	−0.99 ± 0.27 ^ab^	−0.39 ± 0.09 ^ab^	−1.09 ± 0.26 ^b^
	30	−0.44 ± 0.05 ^a^	−0.77 ± 0.17 ^a^	−0.45 ± 0.12 ^a^	−0.15 ± 0.14 ^a^	0.13 ± 0.03 ^a^
b	0	−1.77 ± 0.05 ^a^	−3.01 ± 0.24 ^b^	−3.10 ± 0.32 ^b^	−2.79 ± 0.29 ^b^	−3.26 ± 0.29 ^b^
	15	−3.53 ± 0.28 ^a^	−2.53 ± 0.31 ^a^	−4.37 ± 0.34 ^a^	−3.38 ± 0.23 ^a^	−3.31 ± 0.31 ^a^
	30	−4.19 ± 0.53 ^b^	−1.38 ± 0.24 ^a^	−3.19 ± −0.14 ^b^	−3.24 ± 0.16 ^b^	−2.87 ± 0.38 ^ab^
∆E	0	14.82 ± 0.45 ^a^	11.57 ± 0.99 ^b^	10.70 ± 2.07 ^b^	11.11 ± 1.01 ^b^	6.21 ± 1.48 ^c^
	15	3.95 ± 1.48 ^c^	11.50 ± 2.99 ^a^	2.21 ± 0.48 ^c^	8.11 ± 1.49 ^b^	4.78 ± 0.33 ^c^
	30	3.53 ± 0.94 ^c^	4.86 ± 1.42 ^c^	7.55 ± 0.58 ^a^	6.09 ± 1.60 ^ab^	5.01 ± 1.29 ^bc^
T600 (%)	0	9.18 ± 0.44 ^a^	8.99 ± 0.22 ^b^	8.48 ± 0.25 ^c^	7.76 ± 0.12 ^d^	8.38 ± 0.27 ^c^
	15	9.07 ± 0.25 ^a^	8.73 ± 0.33 ^a^	8.18 ± 0.12 ^b^	7.53 ± 0.17 ^c^	8.01 ± 0.24 ^b^
	30	9.09 ± 0.27 ^a^	8.83 ± 0.42 ^ab^	8.34 ± 0.04 ^b^	7.59 ± 0.45 ^b^	8.26 ± 0.28 ^c^

Different letters in the same column indicate significant differences (*p* < 0.05).

**Table 5 nanomaterials-10-01803-t005:** The models of release behavior and parameters of fitted curves.

Model	Equation	k	n	R^2^
First order	Q = k × (1 − e^−nt^)	0.91202	0.47774	0.94713
Higuchi	Q = k × t^0.5^ + n	0.22132	0.20561	0.897
Korsemeyer-Peppas	Q = k × t^n^	0.49629	0.25826	0.98814

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
