# Peer review of "Long-Term Antibacterial Effect of Electrospun Polyvinyl Alcohol/Polyacrylate Sodium Nanofiber Containing Nisin-Loaded Nanoparticles"

_nanomaterials, 2020, doi:10.3390/nano10091803_

Round 1

Reviewer 1 Report

The authors propose nisin encapsulation nanoparticles on PVA/PAAS nanofibers as a long-term antibacterial agent for water disinfection purposes, thank to the gradual release of nisin. Thus, the PVA/PAAS/EN system shows a very good efficiency as antimicrobial. As the authors recognize, many other alternatives as antimicrobial agents exist. The manuscript should better visualize and critically discuss the advantages / disadvantages of the proposed alternative. Additionally, to the visual check of the antimicrobial effect of the system (Figure 5), the graphs in Figure 5c and 5d are very clear from a quantitative point of view. However, the effect of the ultrasound seems to be not very important. It will be also important to demonstrate the usefulness of the PVA/PAAS/EN system in some real situation. 

Author Response

Comments and Suggestions for Authors 1: The authors propose nisin encapsulation nanoparticles on PVA/PAAS nanofibers as a long-term antibacterial agent for water disinfection purposes, thank to the gradual release of nisin. Thus, the PVA/PAAS/EN system shows a very good efficiency as antimicrobial. As the authors recognize, many other alternatives as antimicrobial agents exist. The manuscript should better visualize and critically discuss the advantages / disadvantages of the proposed alternative. Additionally, to the visual check of the antimicrobial effect of the system (Figure 5), the graphs in Figure 5c and 5d are very clear from a quantitative point of view. However, the effect of the ultrasound seems to be not very important. It will be also important to demonstrate the usefulness of the PVA/PAAS/EN system in some real situation. Re: For the selection of antimicrobial agents, I added the comparison of antibacterial agents in the introduction, which can explain this problem well. You can see it at the beginning of the Introduction. Thank you. Nowadays, microbial pollution has become a major global public health problem. It is an important research direction to add antibacterial agents into basic materials to prepare new materials with antibacterial effect. In the food industry, chemical antimicrobial agents are usually used to inhibit the growth of microorganisms and prolong the shelf life of food. However, in recent years, due to the wide use of chemical antimicrobial agents, food safety problems and human health risks have become a concern (LWT, 2020, 124, 109208). Metal antimicrobial agents (such as silver, copper, zinc, nickel) can show a broad antibacterial spectrum and effectively inhibit microbial growth, but high levels of metal elements significantly increase the risk of poisoning (Colloid. Surface. A., 2020, 602, 125101.). Natural antibacterial agents are the development direction of antibacterial agents in the future. However, essential oils have been widely used as food preservatives. Although essential oils have unique properties, their applications have been limited due to their low antibacterial activity, chemical complexity and strong pungent odor (J. Inorg. Biochem., 2020, 7, 111212.). Bacteriocin has attracted much attention in antimicrobial agents due to its natural, safe and spectrum antibacterial property. As for ultrasound, I think ultrasound is important. The reasons can be divided into the following aspects. First of all, as you said, ultrasound has little effect on antimicrobial activity. Ultrasound does not have the ability to increase or reduce the antibacterial activity, but it can indirectly affect the antibacterial activity. When a large number of nanoparticles are added into the solution, the nanoparticles are the most small independent individual with certain mass, and their distribution in electrospinning is uneven. Therefore, in order to ensure the uniformity of fiber, stirring and ultrasonic are very important. The uneven distribution of nanoparticles in the solution will lead to the uneven distribution of nanoparticles in the fiber. In different ultrasonic time, the antibacterial property of the whole fiber membrane is the same because of the same nanoparticles. However, due to the deposition of nanoparticles in the preparation process, the antibacterial properties of local fibers are different. Secondly, ultrasound can affect the controlled release of nanoparticles. The release of nisin from uniform fibers is uniform and persistent, while the release of non-uniform fibers is uncontrollable. Ultrasound is an important means to control the uniformity of fibers. Furthermore, the nanoparticles which are not uniformly dispersed in the fibers are aggregated and deposited. In the process of absorbing water and releasing the fiber, the nanoparticles gathered inside can not be effectively released or can not be released at all due to the steric hindrance of external nanoparticles, which directly reduces the dosage of effective antibacterial agents. This conclusion can be drawn in Figure 5, although the impact is small, but the impact is real. Some studies have suggested that metal nanoparticles can be well dispersed and uniformly attached to the polymer surface under ultrasonic conditions, which increases the interaction between nanoparticles and polymers. This leads to larger surface area and increases the possibility of contact between nanoparticles and microbial cells, thus enhancing the antibacterial effect (LWT, 2021, 135, 110072.). In addition, it is suggested that the enhancement of antibacterial activity of ultrasonic membrane may be due to the kinetic energy and thermal effect of ultrasonic treatment to promote the release of active substances (Ultrason. Sonochem., 2018, 51, 386-394.). Finally, ultrasonic operation is not only limited to the improvement of antibacterial and controlled-release, but also its influence on the basic properties of the fiber membrane, such as tensile strength, elongation at break and barrier property. It can be seen that the effect of ultrasound on the physical properties of fiber membrane is beneficial and significant. Therefore, ultrasound is very important for the whole experiment. Some studies have shown that the fiber containing nisin nanoparticles can be used in cheese packaging with good results (Lwt-Food Sci. Technol., 2017, 81, 233-242. This study is only for the preparation and characterization of PVA/PAAS/EN nanofibers, and our later experiments will involve its practical application. We also added the expected application direction and the comparison of practical application of the fiber in the full text, such as Line 676. Thank you very much for your comments.

Reviewer 2 Report

The manuscript entitled "Long-term antibacterial effect of electrospun polyvinyl alcohol/polyacrylate sodium nanofibers containing nisin-loaded nanoparticles" is well written and organized, and may be considered for publication after addressing the following recommendations:

Introduction

It is well written and includes a comprehensive review of the literature. However, more emphasis should be paid to the potential application of these nanofibers encapsulating nisin.

Materials and methods

Lines 104-112: The encapsulation of nisin was performed through a freeze drying process of a double emulsion. Why did the authors select this double emulsion system? Would it be possible to perform the encapsulation with a single emulsion system or directly blending the bioactive with the polymers of the nanofibers? What was the reason to perform a freeze drying process instead any other drying method? After the ultrasonication step, the authors perform a centrifugation step, what is the objective of this step? This centrifugation step, did not affect the stability of the emulsion?

Lines 150-153: Was the electrospinning equipment commercial or home-made? Was the process carried out at ambient conditions, or did you have controlled these parameters?

Line 164: The equations must be numbered, and the meaning of each variable must be described.

Lines 174: It would be better to indicate the meaning of the abbreviation of SR

Lines 182-188: It is supposed that the CIELAB method was used for the color determination, but it should be clarified in this section. The meaning of each term in the equation should also be clarified.

Lines 204-205: The methodology of the antimicrobial experiment should be summarized even if the authors have followed the methodology descripted in reference 23.

Results and discussion

Section 3.2: The authors report the encapsulation efficiency and the particle size, it would be interesting to report the loading capacity of the microparticles as well as the nisin loading capacity in the fibers.

Section 3.3: In the reported images, particles seemed to be agglomerated, could this be due to the freeze drying process? It seems that this agglomeration affects also the nanofiber morphology, if other drying method were used, do you expect to have similar results?

Section 3.4: Why were the spectra of the encapsulates not a combination of the spectra of the gelatin and nisin? Was the soy oil also entrapped inside of the capsule? It would be also interesting to include the spectra of the PVA and PAAS alone.

Supplementary material

Check the values reported in Table S1, since they are not the same as the values reported throughout the text.

Several spelling mistakes were detected throughout the text:

                One a is missing in the title in polyvinyl alcohol

                Line 22 in the abstract should say Korsemeyer-Peppas model

                Line 38 in the Introduction g L-1, -1 should be a superscript. All the units should be revised throughout the text in the same way.

Author Response

Comments and Suggestions for Authors 2:

The manuscript entitled "Long-term antibacterial effect of electrospun polyvinyl alcohol/polyacrylate sodium nanofibers containing nisin-loaded nanoparticles" is well written and organized, and may be considered for publication after addressing the following recommendations:

  1. Introduction: It is well written and includes a comprehensive review of the literature. However, more emphasis should be paid to the potential application of these nanofibers encapsulating nisin.

Re: In the previous studies, nisin nanoparticle nanofibers were mainly prepared by adding nisin nanoparticles into electrospinning solution. Its main application scope is to wrap such nanofibers in solid food, such as cheese. I added the scope of application in the Introduction, as follows, thanks. Soto et al. prepared biodegradable antimicrobial nanofibers based on amaranth protein isolate: pullulan (API: PUL) plus nisin (Int. J. Food. Microbol., 2019, 295, 25-32.). The nisin in the (API: PUL) nanofibers had a release rate of 81.49% at pH 3.4 after 12 h. When nisin API:PUL fibers were applied to complete bactericidal activity against Salmonella Typhimurium, L. monocytogenes and L. mesenteroides inoculated in fresh cheese, microorganism inactivation was complete after 142, 120 and 170 h, respectively. Cui et al. reported that polyethylene oxide nanofibers embedded with nisin-loaded poly-γ-glutamic acid/chitosan nanoparticles were highly efficacious against L. monocytogenes. The L. monocytogenes density on fresh cheesed ecreased from 3.19 log CFU g-1 to 1.43 log CFU g-1 after 7 d (Lwt-Food Sci. Technol., 2017, 81, 233-242).

  1. Materials and methods: Lines 104-112: The encapsulation of nisin was performed through a freeze drying process of a double emulsion. Why did the authors select this double emulsion system? Would it be possible to perform the encapsulation with a single emulsion system or directly blending the bioactive with the polymers of the nanofibers? What was the reason to perform a freeze drying process instead any other drying method? After the ultrasonication step, the authors perform a centrifugation step, what is the objective of this step? This centrifugation step, did not affect the stability of the emulsion?

Re: For the discussion of single phase emulsion and biphasic emulsion, I must point out that the potential applications of biphasic emulsion are numerous, and the research of these systems is now an active research field. This is especially true in product areas such as drug delivery systems, cosmetics and food. Double emulsions are considered extremely promising formulations for slow and controlled release of entrapped active matter from the inner phase to the outer continuous phase. This function can mask the taste and or odor and prevent oxidation, light or enzymatic degradation. It also provides controlled release of the active ingredient, which is triggered by dilution, shearing or agitation. This is more stable and intelligent for nisin control release (In Interface Science and Technology, 2004.).

First of all, the antibacterial activity of nisin is usually affected by environmental factors, such as pH, temperature and food matrix (such as proteins, lipids and enzymes). Nanoparticle encapsulation can solve this problem well, such as liposomes, alginate, chitosan and other nanoparticle embedding systems (Food Control., 2017, 79, 317-324.). The original intention of this research is to fully protect nisin and give full play to its antibacterial effect. Secondly, nisin can be directly added to the electrospinning solution, but nisin is low water solubility, so if it is added to the electrospinning solution, a solvent that can dissolve the electrospinning substrate and nisin at the same time is required (Food Control., 2012, 24, 184-190.). I believe that there are not many choices. Not only did it fail to expand the scope of application of nisin, but the solvent may cause the antibacterial effect of nisin. Furthermore, if it is directly added to the electrospinning solution, the nisin is miscible with the substrate and cannot be fully embedded, and the environmental impact on the nisin increases, which is unfavorable to the antibacterial activity and application scope of the nisin.

There are many drying methods for nanoparticles, including freeze-drying, ordinary drying, etc. Freeze drying is a very common method for microencapsulation drying. It can protect nisin to the maximum extent and avoid the influence of environment when operating in vacuum and low temperature when operated under vacuum and low temperature. The double emulsion system is easy to separate into single emulsion, and freeze-drying is easy to maintain double emulsion stability. The lyophilized nanoparticles have fast rehydration and easy internal nisin controlled release (Food Chem., 2019, 300, 125171.).

Due to the influence of the preparation method of nanoparticles, the prepared solutions are not all nanoparticles. The nanoparticles embedded nisin were deposited at the bottom, and the rest were not target substances, but they accounted for most of the volume and needed to be removed. Centrifugation is a process of enrichment, which can effectively separate nanoparticles and useless substances, which also lays the foundation for improving the freeze-drying rate. Centrifugation and freeze-drying are two inseparable and common steps in the preparation of nanoparticles.

Centrifugation does not change the stability of nanoparticles. Firstly, in the process of preparation of nanoparticles, the nanoparticles are stable and stable, which can be observed in SEM images. Secondly, the centrifugation process is only a process to accelerate the physical deposition of the nanoparticles, and does not affect the stability of the nanoparticles. Moreover, if the physical deposition process will affect the stability of the nanoparticles, then the nanoparticle research is a failure. Moreover, if the centrifugation process will affect the stability of nanoparticles, only those nanoparticles with unstable inclusion will be affected, which is also a screening process for nanoparticles. This is very common in many studies of nanoparticle preparation (Colloid. Surface. A., 2020, 605, 125354.).

  1. Lines 150-153: Was the electrospinning equipment commercial or home-made? Was the process carried out at ambient conditions, or did you have controlled these parameters?

Re: The electrospinning equipment we used was commercial. The model and manufacturer of the machine have been added. And we controlled these parameters in the electrospinning process. Thanks.

  1. Line 164: The equations must be numbered, and the meaning of each variable must be described.

Re: Corrected. Thanks.

  1. Lines 174: It would be better to indicate the meaning of the abbreviation of SR

Re: Corrected. Thanks.

  1. Lines 182-188: It is supposed that the CIELAB method was used for the color determination, but it should be clarified in this section. The meaning of each term in the equation should also be clarified.

Re: It had been clarified. Thanks.

  1. Lines 204-205: The methodology of the antimicrobial experiment should be summarized even if the authors have followed the methodology descripted in reference 23.

Re: Corrected. Thanks. The antibacterial properties of nanofibers were measured using the previous method. The antibacterial activity of nanofiber membrane was evaluated by agar diffusion method. The target strain was inoculated in the ordinary nutrient liquid culture medium, and was cultured at 37 ℃ for 24 h, and then continuously diluted to 10-7 CFU/ ML. 0.10 mL bacterial cells were coated on the surface of tryptone soybean agar (TSA) medium, and three nanofiber discs were placed on the inoculation surface. The plate was cultured at 37 ℃ for 24 hours, and the inhibition zone was measured by digital caliper. The antibacterial test was divided into 3 times and the average value was taken (Int. J. Biol. Macromol., 2019, 141, 378-386).

  1. Results and discussion: Section 3.2: The authors report the encapsulation efficiency and the particle size, it would be interesting to report the loading capacity of the microparticles as well as the nisin loading capacity in the fibers.

Re: Sorry, we didn't test the load capacity of particles and nisin. Because of our electrospinning process, as you know, we add a fixed amount of nanoparticles into the same electrospinning solution to prepare a fixed concentration solution. In the process of spinning the solution into fiber membrane, we control the amount of solution, flow rate, voltage, etc. The content of nanoparticles in each fiber membrane prepared by us is the content set by us, but we do not characterize the number of nanoparticles tested. We characterize the loading capacity of nanoparticles in the fiber by quality. Obviously, they are the same. Similarly, with the same nanoparticle mass and similar antibacterial data, it can be seen that the loading capacity of nisin in fiber is the same. Therefore, it is unnecessary to test the loading capacity of particles and nisin in fibers. In other people’s studies, these two were not tested (LWT., 2017, 81, 233-242.). We will consider and adopt your suggestion in future research. Thank you very much for your suggestion.

  1. Section 3.3: In the reported images, particles seemed to be agglomerated, could this be due to the freeze drying process? It seems that this agglomeration affects also the nanofiber morphology, if other drying method were used, do you expect to have similar results?

Re: The research of Calderón-Oliver et al. (Figure 1) shows that the nanoparticles obtained by freeze drying method are amorphous, while the nanoparticles obtained by spray drying method have different hemispheres. Freeze dried coacervation layer shows spongy structure, especially empty particles (Fig. 6a and b). This spongy structure will disappear with the addition of core material (Fig. 6c and d) (Food Hydrocolloid., 2017, 62, 49-57.). However, in the study of Hu et al., (Figure 2) the particles prepared by freeze-drying were similar to those in this study and similar to the spray dried particles in Figure 1 (Food Control., 2017, 43-52.). Therefore, we can infer that the drying method will not affect the morphology of particles. Aggregation is a normal phenomenon, and most people's articles also show aggregation. Because the size of EN particles is too small, no matter which drying method is used, each particle can not be absolutely separated. At this time, when it is added into the electrospinning solution, the effect of ultrasonic is particularly important. The cavity effect produced by ultrasonic wave is the rupture of microbubbles, which breaks and propagates through the solution in the form of sound wave. Therefore, ultrasound can make the particles evenly distributed in the solution on the macro scale, while the cavitation effect can make the particles aggregate smaller and more uniform at the micro scale (Ultrason. Sonochem., 2019, 59, 104731.).

  1. Section 3.4: Why were the spectra of the encapsulates not a combination of the spectra of the gelatin and nisin? Was the soy oil also entrapped inside of the capsule? It would be also interesting to include the spectra of the PVA and PAAS alone.

Re:. Because the manufacturing process of EN is not a simple physical blending, it will not show a completely consistent binding phenomenon. At the same time, nisin may affect the intermolecular conjugation of EN, which changes the vibration of the original functional groups at 3401 cm-1 and 2357 cm-1, resulting in image difference. This can also be observed in the study of nisin nanoparticle preparation (Food Control., 2017, 79, 317-324.).

Oil does remain in the nanoparticles, which are encapsulated as oil phase, but in very small amounts. Moreover, the oil exists in nanoparticles. We provide FTIR images of nanoparticles with different concentrations. Comparing the two images, it can be seen that the oil residue does not affect the FTIR images and functional groups. We have added separate PVA and PAAS FTIR images.

According to your opinion, we have added FTIR images of PVA and PVA / PAAS. Since we have provided FTIR images of PVA/PAAS/EN, we can analyze the influence of adding PAAS and EN on the functional groups of fiber membrane layer by layer after adding, so that the organization will be clearer. Thank you.

  1. Supplementary material: Check the values reported in Table S1, since they are not the same as the values reported throughout the text.

Re: The data in Appendix S1 is the boundary I set, while the data in Table 2 and the data mentioned later are excellent combination conditions calculated and screened by response surface methodology. You can see it in the previous literature (Chem. Eng. J. Advances, 2020, 26, 100007).

  1. Several spelling mistakes were detected throughout the text: One a is missing in the title in polyvinyl alcohol

Re: Corrected. Thanks.

  1. Line 22 in the abstract should say Korsemeyer-Peppas model

Re: Done, thanks.

  1. Line 38 in the Introduction g L-1, -1 should be a superscript. All the units should be revised throughout the text in the same way.

Re: We had gone through the whole manuscript and corrected them. Thanks.

Reviewer 3 Report

This manuscript describes comprehensive materials properties of electrospun polyvinyl alcohol/polyacrylate sodium nanofibers with nisin. The manuscript dealt with a large set of data and found the optimal conditions. I would recommend the acceptance with major revision of manuscript based on several comments below.

  1. Line 3: Missing letter in title – “lcohol/polyacrylate”
  2. Line 86, “nisin encapsulation (EN)”: The abbreviation seems wrong to me. Author used “EN” in entire manuscript and I suggest “encapsulated nisin (EN)”.
  3. Line 327-328: Author describes the fiber diameter increase can be explained by crosslinking of EN. Can EN has enough density to cover entire fibers uniformly? Fibers in figure 2(i) seems highly homogeneous.
  4. Line 329, “overall mechanical properties”: Overall is too broad. Can you specify properties?
  5. Line 330, “nanofibers were disrupted”: Figure 2(j) shows one (or maybe two) broken nanofibers. I suggest authors add the quantitative image analysis of pores of nanofibers after sonication.
  6. Line 422, “ultrasonication also roughened the nanofiber surface”: Figure 2(j) (30 min sonication) does not clearly show fiber surface roughness. Fiber surface in figure 2(h) (0 min sonication) seems rougher than 2(j). Please confirm this.
  7. Line 458, “Nanofiber color”: I suggest visual data (photograph etc.) may be better to readers to see color changes.
  8. Figure 4 seems very busy. It contains data for the section 3.11 and 3.12 together. I would suggest divide in two figures.
  9. Line 526: Missing letter in “lcohol/polyacrylate”
  10. Line 565, 570, “NA+”: -> Na+

Author Response

Comments and Suggestions for Authors 3:

This manuscript describes comprehensive materials properties of electrospun polyvinyl alcohol/polyacrylate sodium nanofibers with nisin. The manuscript dealt with a large set of data and found the optimal conditions. I would recommend the acceptance with major revision of manuscript based on several comments below.

  1. Line 3: Missing letter in title – “lcohol/polyacrylate”

Re: Corrected. Thanks.

  1. Line 86, “nisin encapsulation (EN)”: The abbreviation seems wrong to me. Author used “EN” in entire manuscript and I suggest “encapsulated nisin (EN)”.

Re: We had decided to follow your comment and correct. Thanks.

  1. Line 327-328: Author describes the fiber diameter increase can be explained by crosslinking of EN. Can EN has enough density to cover entire fibers uniformly? Fibers in figure 2(i) seems highly homogeneous.

Re: Nisin particles can not completely cover the fiber membrane. It can be seen from the figure that the fiber diameter is increased at the existing concentration. Secondly, the fiber diameter increases with the increase of additives, which is very common in the study of electrospun films (Ultrason. Sonochem., 2019, 59, 104731.). Because 2i is the best group for me to undergo ultrasonic treatment, and the nanoparticles are evenly distributed in the fiber, so it looks smoother than the group without ultrasonic treatment. Moreover, the coarseness of the fibers in the ultrasound free group may be due to the aggregation of nanoparticles.

  1. Line 329, “overall mechanical properties”: Overall is too broad. Can you specify properties?

Re: Overall mechanical properties means TS and EAB. I have revised it in the article. Thank you.

  1. Line 330, “nanofibers were disrupted”: Figure 2(j) shows one (or maybe two) broken nanofibers. I suggest authors add the quantitative image analysis of pores of nanofibers after sonication.

Re: The higher the density is, the lower the porosity is. If only one of them is studied, the distribution of fiber can be explained. The results show that the density of the fiber increases first and then decreases with the increase of ultrasonic time. The qualitative and quantitative data are complete, so it is unnecessary to study porosity, and only one of them has been studied in many studies (Ultrason. Sonochem., 2019, 59, 104731.).

  1. Line 422, “ultrasonication also roughened the nanofiber surface”: Figure 2(j) (30 min sonication) does not clearly show fiber surface roughness. Fiber surface in figure 2(h) (0 min sonication) seems rougher than 2(j). Please confirm this.

Re: Thank you for asking this question. It's true that I made a mistake and I have already revised it. Thanks.

  1. Line 458, “Nanofiber color”: I suggest visual data (photograph etc.) may be better to readers to see color changes.

Re: The electrospun film prepared by PVA is white, and the nanoparticles are also white. Although the difference is not as intuitive as the picture, the color difference is less than the threshold value that can be distinguished by human eyes, so it is difficult to see the difference by picture observation. This phenomenon can be seen from Table 4, with the increase of nanoparticle concentration, the color does not change too much.

  1. Figure 4 seems very busy. It contains data for the section 3.11 and 3.12 together. I would suggest divide in two figures.

Re: It's separated. Thank you.

  1. Line 526: Missing letter in “lcohol/polyacrylate”

Re: Corrected. Thanks.

  1. Line 565, 570, “NA+”: -> Na+

Re: Corrected. Thanks.

Reviewer 4 Report

Line 14 abstract I recommend to slightly change the starting of the manuscript to passive voice. What is the meaning of "good morphology". 

In the introduction, it is very important that the authors described clearly the advantages of the polymers that they have processed and the possible applications. They mentioned that they "optimized the amount of niosin", however, Han et al (Acta Biomaterialia 2017) reported encapsulation efficiencies of 95 and 87%. Then, I do not understand what is the optimization. The authors should be clear because the introduction is also very confusing with many abbreviations. 

In some parts, the authors mentioned that the have used previously methodologies. I recommend writing the procedure or adding the reference because they do both and is confusing (line 106). In this case, should the reader go to the second author and try that methodology or refer to the current manuscript. 

What is the meaning of agar diffusion test. I guess that the authors are determining the activity but not the concentration because the method is qualitative and not quantitative.Please, explain. 

  Should the reader go to the article and read the methodology. In this case, what is the mistake of the determination. It is confusing. What is the difference between that method and the BCA kit.

Table 2, the letters should be changed because it is very confusing for the reader to identify the mistake by a,b,c,

what is the meaning of unlimited ultrasonication time? (line 275)

Sorry, but what is the meaning of when the ultrasonication time exceeded a certain duration? hours, minutes, days...(line 282)

line 283 embed is wrong the past particle is embedded 

About the references, please, try to organize the references and explain the advantages of electrospinning, the used polymers for the encapsulation of niosin. Sorry, but the introduction doesnot explain the novelty. 

Author Response

Comments and Suggestions for Authors 4:

  1. Line 14 abstract I recommend to slightly change the starting of the manuscript to passive voice. What is the meaning of "good morphology".

Re: It means that the film has a good morphology, that is, the fiber surface is smooth, no fracture, and the diameter is uniform. This description is not only in this study, but also in other studies (LWT, 2019, 113, 108292.). Thanks.

  1. In the introduction, it is very important that the authors described clearly the advantages of the polymers that they have processed and the possible applications. They mentioned that they "optimized the amount of niosin", however, Han et al (Acta Biomaterialia 2017) reported encapsulation efficiencies of 95 and 87%. Then, I do not understand what is the optimization. The authors should be clear because the introduction is also very confusing with many abbreviations.

Re: We have added the advantages of polymers and their application scope in the Introduction. In the study, they did not optimize the content of nisin, but fixed the content of nisin, and compared the entrapment efficiency of coaxial electrospinning and triaxial electrospinning. They just compared to get a better spinning method, and did not optimize the parameters (Acta Biomater., 2017, 53, 242-249.). If you have any questions, you can communicate with us. Thank you.

  1. In some parts, the authors mentioned that the have used previously methodologies. I recommend writing the procedure or adding the reference because they do both and is confusing (line 106). In this case, should the reader go to the second author and try that methodology or refer to the current manuscript.

Re: Corrected. Thanks.

  1. What is the meaning of agar diffusion test. I guess that the authors are determining the activity but not the concentration because the method is qualitative and not quantitative. Please, explain.

Re: It can be measured quantitatively. The antibacterial activity of nisin was measured by agar diffusion experiment. The standard curve of nisin concentration was drawn. Then the nisin concentration in the experiment was obtained by agar diffusion experiment.

  1. Should the reader go to the article and read the methodology. In this case, what is the mistake of the determination. It is confusing. What is the difference between that method and the BCA kit.

Re: When consulting the literature, we found that most people used agar diffusion method, so we chose this method. It is undeniable that BCA kit is a rapid method to detect protein content, which can reduce a lot of work. However, the principle of this method is the same as that of the BCA kit, and the method and results are correct and scientific. I will consider your opinion in the future research. Thank you for your suggestion (Mat. Sci. Eng. C., 2017, 76, 673-683.).

  1. Table 2, the letters should be changed because it is very confusing for the reader to identify the mistake by a, b, c,

Re: I think there is no problem with the significant expression letters in Table 2. Do you mean Table 4? I will change the color expression letters L, a, b in Table 4 to L*, a*, b*, which will not cause misunderstanding. Thank you.

  1. what is the meaning of unlimited ultrasonication time? (line 275)

Re: EE increased with ultrasonication time when the time does not reach 3 min. I have modified it, thanks.

  1. Sorry, but what is the meaning of when the ultrasonication time exceeded a certain duration? hours, minutes, days. (line 282)

Re: The certain duration is the optimal time of 3 min. In order to avoid misunderstanding, I have revised it. Thank you.

  1. line 283 embed is wrong the past particle is embedded

Re: Corrected. Thanks.

  1. About the references, please, try to organize the references and explain the advantages of electrospinning, the used polymers for the encapsulation of niosin. Sorry, but the introduction doesnot explain the novelty.

Re: Added, thanks. Gelatin is a biodegradable protein material with excellent water solubility, emulsification and thickening capabilities, and high crosslinking activity. It is produced by partial hydrolysis of collagen, which is still the main commercial choice for wall materials (Int. J. Biol. Macromol., 2019, 133, 722-731.). De Souza et al. studied gelatin and five different polysaccharides, such as Arabic gum and pectin, to encapsulate proanthocyanidins rich cinnamon extract by composite coacervation. The particles obtained from different materials have high entrapment efficiency. The microencapsulation process maintains the bioactivity potential of cinnamon extract and conceals the undesirable sensory properties, so that it can be used as a functional component in food and as a health care product (Food Hydrocolloid., 2017, 77, 297-306.). In the work of Oliveira et al., the green coffee oil loaded with caffeine and kawasol was encapsulated with cashew gum and gelatin to prepare nanoparticles. The particle with 25% green coffee oil had good encapsulation efficiency (85.57%). The nanoparticles were stable under the processing conditions of tamarind juice, which were allowed to be mixed into the juice without changing its rheological or sensory properties, and remained stable during storage for 30 days (Food Res. Int., 2020, 131, 109047.). The coacervates were able to encapsulate the lipid extract (astaxanthin encapsulation efficiency 59.9 ± 0.01%), forming multinucleated, polymorphic nanoparticles with an average size of 32.7 ± 9.7 μm by gelatin and cashew gum. Nanoparticles are well dispersed in pure yoghurt, and it can improve the coloring ability, although no differences in odor were found (Food Hydrocolloid. Oxford., 2016, 61, 155-162.).

Several studies reported that the nanoparticles prepared by gelatin encapsulation were successfully applied in food industry. The microencapsulation of nisin prepared by gelatin encapsulation can effectively protect nisin and has broad development prospects in food industry.

Polymer based nanofibers are considered as potential materials in a wide range of fields due to their excellent properties, such as high specific surface area and easy functionalization. Electrospinning technology, including multi axis electrospinning, is a general method for the production of fiber membranes of various natural and synthetic materials (Acta Biomater., 2017, 53, 242-249.). Electrospinning has been investigated for protective textiles, biomedical and food packaging as it generates nanofibers with properties not found in traditional fibers such as high pore interconnectivity, high specific surface area, surface functionalization, excellent breathability and tunable porosity and easy manipulation of chemical compositions and structures for desired properties and functionalities (Colloid. Surface B., 2016, 146, 144-151.).

Round 2

Reviewer 1 Report

The authors have produced an improved version of the manuscript. New comments and information have been included in the revised version. The role of ultrasound has been clarified. The preparation and characterization of PVA/PAAS/EN nanofibers is of interest, but its usefulness in real situations is the final demonstration of its achievement, especially in this type of work with a practical interest, which justify the previous work. Although preliminary tests should be included in the present manuscript.

Author Response

Comments and Suggestions for Authors 1:

The authors have produced an improved version of the manuscript. New comments and information have been included in the revised version. The role of ultrasound has been clarified. The preparation and characterization of PVA/PAAS/EN nanofibers is of interest, but its usefulness in real situations is the final demonstration of its achievement, especially in this type of work with a practical interest, which justify the previous work. Although preliminary tests should be included in the present manuscript.

Re: Thanks for your valuable comments. Practical application has a very important role in the assessment of nanofiber characterization, but because of the layout and time constraints, this paper proposed to study the preparation and characterization of PVA/PAAS/EN nanofibers only. Actually, we have prepared more in-depth study of practical application to evaluate the effect and mechanisms of PVA/PAAS/EN nanofibers on the preservation of different kinds of food. And the research work will be completed later. Given your interest in practical application, part of experimental results and findings of other scholars are presented in the present manuscript.

Reviewer 3 Report

I would recommend the acceptance of current revised manuscript for the publication in Nanomaterials. 

Author Response

I would recommend the acceptance of current revised manuscript for the publication in Nanomaterials.

Re: Thanks.

Reviewer 4 Report

The authors have to improve the quality of the figures. Example fig. 1 is impossible to read the characters. Figure 3 is so long and confusing. This figure has to be abstracted to give important information. Figure 5 is also not clear.

Line 489 does not belong to swelling studies. 

The English of this manuscript has to be improved as several mistakes are still present in the text. 

Author Response

Comments and Suggestions for Authors 3:

The authors have to improve the quality of the figures. Example fig. 1 is impossible to read the characters. Figure 3 is so long and confusing. This figure has to be abstracted to give important information. Figure 5 is also not clear.

Re: Corrected. Thanks.

Line 489 does not belong to swelling studies.

Re: Corrected. Thanks.

The English of this manuscript has to be improved as several mistakes are still present in the text.

Re: Corrected. Thanks.